https://doi.org/10.1038/s41467-019-12126-4　　**OPEN**

# TRIM66 reads unmodified H3R2K4 and H3K56ac to respond to DNA damage in embryonic stem cells

Jiajing Chen[1,4], Zikang Wang[2,4], Xudong Guo [2,4], Fudong Li[1], Qingtao Wei [1], Xuwen Chen[1], Deshun Gong [3], Yanxin Xu[2], Wen Chen[2], Yongrui Liu[1], Jiuhong Kang[2] & Yunyu Shi[1]

Recognition of specific chromatin modifications by distinct structural domains within "reader" proteins plays a critical role in the maintenance of genomic stability. However, the specific mechanisms involved in this process remain unclear. Here we report that the PHD-Bromo tandem domain of tripartite motif-containing 66 (TRIM66) recognizes the unmodified H3R2-H3K4 and acetylated H3K56. The aberrant deletion of Trim66 results in severe DNA damage and genomic instability in embryonic stem cells (ESCs). Moreover, we find that the recognition of histone modification by TRIM66 is critical for DNA damage repair (DDR) in ESCs. TRIM66 recruits Sirt6 to deacetylate H3K56ac, negatively regulating the level of H3K56ac and facilitating the initiation of DDR. Importantly, Trim66-deficient blastocysts also exhibit higher levels of H3K56ac and DNA damage. Collectively, the present findings indicate the vital role of TRIM66 in DDR in ESCs, establishing the relationship between histone readers and maintenance of genomic stability.

[1] Hefei National Laboratory for Physical Sciences at Microscale and School of Life Sciences, University of Science and Technology of China, Hefei, Anhui 230026, China. [2] Clinical and Translational Research Center of Shanghai First Maternity and Infant Hospital, Shanghai Key Laboratory of Signaling and Disease Research, Collaborative Innovation Center for Brain Science, School of Life Sciences and Technology, Institute for Advanced Study, Tongji University, Shanghai 200092, China. [3] Beijing Advanced Innovation Center for Structural Biology, Tsinghua-Peking Joint Center for Life Sciences, School of Life Sciences, Tsinghua University, Beijing 100084, China. [4] These authors contributed equally: Jiajing Chen, Zikang Wang, Xudong Guo. Correspondence and requests for materials should be addressed to J.K. (email: jhkang@tongji.edu.cn) or to Y.S. (email: yyshi@ustc.edu.cn)

TRIM family proteins are characterized by the tripartite motif (TRIM), which contains one RING-finger domain, one or two B-box domains, and a coiled-coil region at the N terminus[1]. TRIM24, TRIM28, and TRIM33 are in the C-VI subfamily of the TRIM family, owing to their conserved plant homeodomain (PHD)-Bromodomains at the C terminus. Both PHD finger and Bromodomain are evolutionarily and structurally conserved modules. The majority of canonical single PHD fingers are reported to read the N-terminal tail of histone H3, mainly the methylation status of H3K4 (H3K4me2/3 or H3K4me0)[2], whereas the Bromodomains are able to read the acetylated lysine, especially that on histone H3 and H4[3–12]. In recent years, the structure and function of the PHD Bromodomains of TRIM24, TRIM28, and TRIM33 have been studied extensively[5,6,13]. Both of the PHD-Bromodomains of TRIM24 and TRIM33 read post-translational modifications (PTMs) in a combinatorial manner, which might improve the binding affinity and specificity of the interaction, thus resulting in diverse downstream outcomes[5,6,14]. The paralog of TRIM24, TRIM28, and TRIM33—termed TRIM66—also contains a PHD-Bromodomain at its C terminus. Early studies reported that the expression of TRIM66 was largely restricted to the testes at the early elongating spermatid stage. In addition, TRIM66 exerted a deacetylase-dependent silencing effect in mammalian cells and interacted with heterochromatin protein 1[15]. However, little is known about the structure and function of the TRIM66 PHD-Bromodomain and it remains a task to figure out whether the PHD-Bromodomain of TRIM66 plays a critical role in its cellular function.

The maintenance of genomic stability is vital for safeguarding the complete inheritance of genetic material in organisms. Internal and external stimuli, including DNA replication, cell metabolism, and environmental toxins may persistently damage the genomic DNA, leading to premature aging, apoptosis, or tumorigenesis. Embryonic stem cells (ESCs) are capable of self-renewal and differentiation into all cell types, playing important roles in the development and clinical application[16,17]. ESCs possess a more vigorous self-renewal and metabolism process compared with somatic cells, and are exposed to a higher risk of DNA damage. However, they exhibit a lower mutation rate[18–20], suggesting the presence of a more efficient DNA repair mechanism in ESCs. Reportedly, ESCs fail to activate the G1-S and intra-S checkpoints in response to DNA lesions. However, they can be temporarily arrested in the G2 phase following ataxia telangiectasia mutated (ATM) activation[21–23]. In addition, specific regulators of DNA repair exist in ESCs. Spalt-like transcription factor 4, favoring ATM activation by interacting with RAD50, regulates the DNA damage response[24]. The FILIA protein safeguards genomic stability through promoting poly [ADP-ribose] polymerase 1 activity in mouse ESCs[25]. However, the key molecules and mechanisms involved in the efficient DNA damage repair (DDR) of ESCs remains to be fully elucidated.

The diverse types of histone PTMs, such as acetylation, methylation, and phosphorylation, are crucial for eukaryotic epigenetic regulation[14]. The dynamic acetylation of histone lysine alters the accessibility of chromatin and affects the recruitment of non-histone proteins, which are closely engaged in DDR and maintenance of genomic integrity[26–28]. The H3K56 is on the lateral surface of the nucleosome, which is close to the DNA entry/exit site and interacts with DNA[29]. The H3K56ac occurs on the core histone and is mainly catalyzed on free and newly synthesized histones. These are subsequently used to assemble the nucleosomes[29]. Early studies reported that the H3K56ac influences the assembly or disassembly of the nucleosome during DNA repair, replication, or transcription in yeast[30,31]. In mammals, the dynamic change of H3K56ac, which is rapidly deacetylated at the DNA double-stranded break (DSB) sites and

restored at the late stage of DNA repair, is necessary for the regulation of DDR[32–36]. Recently, H3K56ac was also reported to be involved in the regulation network of ESC pluripotency[37–39]. However, the specific reader for H3K56ac and its exact mechanisms in the DDR of ESCs remains unclear.

Herein, we determine the crystal structure of the TRIM66 PHD-Bromodomain in its free state and in a complex with the N terminus of histone H3. Our results show that the PHD domain of TRIM66 recognizes both unmodified H3R2 and H3K4 residues, whereas the Bromodomain interacts with the H3K56ac peptide through its ZA and BC loops. Functional analysis show that loss of TRIM66 lead to severe DNA damage, genomic instability, and retention of H3K56ac following DNA damage in ESCs. Further studies show that TRIM66 regulates the deacetylation of H3K56ac by recruiting histone deacetylase Sirt6, providing the appropriate chromatin environment for subsequent DNA repair and safeguarding the genomic integrity of ESCs.

## Results

**TRIM66 PHD-Bromo binds unmodified H3 N-terminal and H3K56ac.** TRIM66 has a similar domain organization (Fig. 1a) with TRIM24, TRIM28, and TRIM33[15]. The sequences of the PHD-Bromodomains are highly conserved among TRIM24, TRIM33, and TRIM66. The sequence alignment of the PHD-Bromodomains of TRIM66 from different species also demonstrates high conservation (Supplementary Fig. 1a, b).

We first determined the crystal structure of the TRIM66 PHD-Bromodomain. We used the mutant TRIM66-MUT$_{968-1160}$ protein (Supplementary Fig. 1c–e, Supplementary Table 1, and Supplementary Note 1) for the crystallization experiment and determined the crystal structure of the TRIM66 PHD-Bromodomain in free state (Table 1). The 2.1 Å crystal structure of the TRIM66 PHD-Bromodomain showed that the PHD domain adopted a typical cross-braced topology, whereas the Bromodomain adopted a typical left-handed four-helical bundle topology. There was an extensive interface between the PHD and Bromodomain, enabling the formation of a unified structural unit (Fig. 1b). The structure of PHD-Bromodomain of TRIM66 was similar to that of TRIM24 and TRIM33 (Fig. 1c).

We next performed the glutathione S-transferase (GST) pulldown assays of the TRIM66-WT$_{965-1160}$ against the calf thymus histones to verify the histone-binding ability of TRIM66 PHD-Bromodomain. The results showed that TRIM66-WT$_{965-1160}$ could interact with histones, primarily H3 and H4 (Supplementary Fig. 2a). The PHD of TRIM66 shared a low conservation with those of ING4, PHF2, and BPTF, and lacked the conserved aromatic residues, which were necessary for binding H3K4me2/3[40–42] (Supplementary Fig. 2b). Nevertheless, TRIM66 was highly conserved in sequence and structure with the PHDs of TRIM24 and TRIM33 (Fig. 1c and Supplementary Fig. 2b), which were reported to bind H3K4me0[5,6]; thus, we speculated that the PHD of TRIM66 could bind H3K4me0. To verify this hypothesis, we performed isothermal titration calorimetry (ITC)-based binding assays. The results showed that the TRIM66-WT$_{965-1160}$ bound unmodified H3$_{1-12}$ with a dissociation constant ($K_D$) of 15.7 μM (Fig. 1d). The sequence alignment and structure comparison of the Bromodomain of TRIM66 with TRIM24, TRIM33, CBP, BPTF, and GCN5 indicated that the overall fold of TRIM66 Bromodomain was highly conserved and it may also be an acetyl-lysine reader (Fig. 1c and Supplementary Fig. 2c, d). Based on previous researches reporting the acetyl-lysine binding partners of Bromodomain[3–12], we chose a series of related peptides derived from H3 and H4, to test their interactions with TRIM66-WT$_{965-1160}$ by ITC experiments. No obvious bindings were observed between

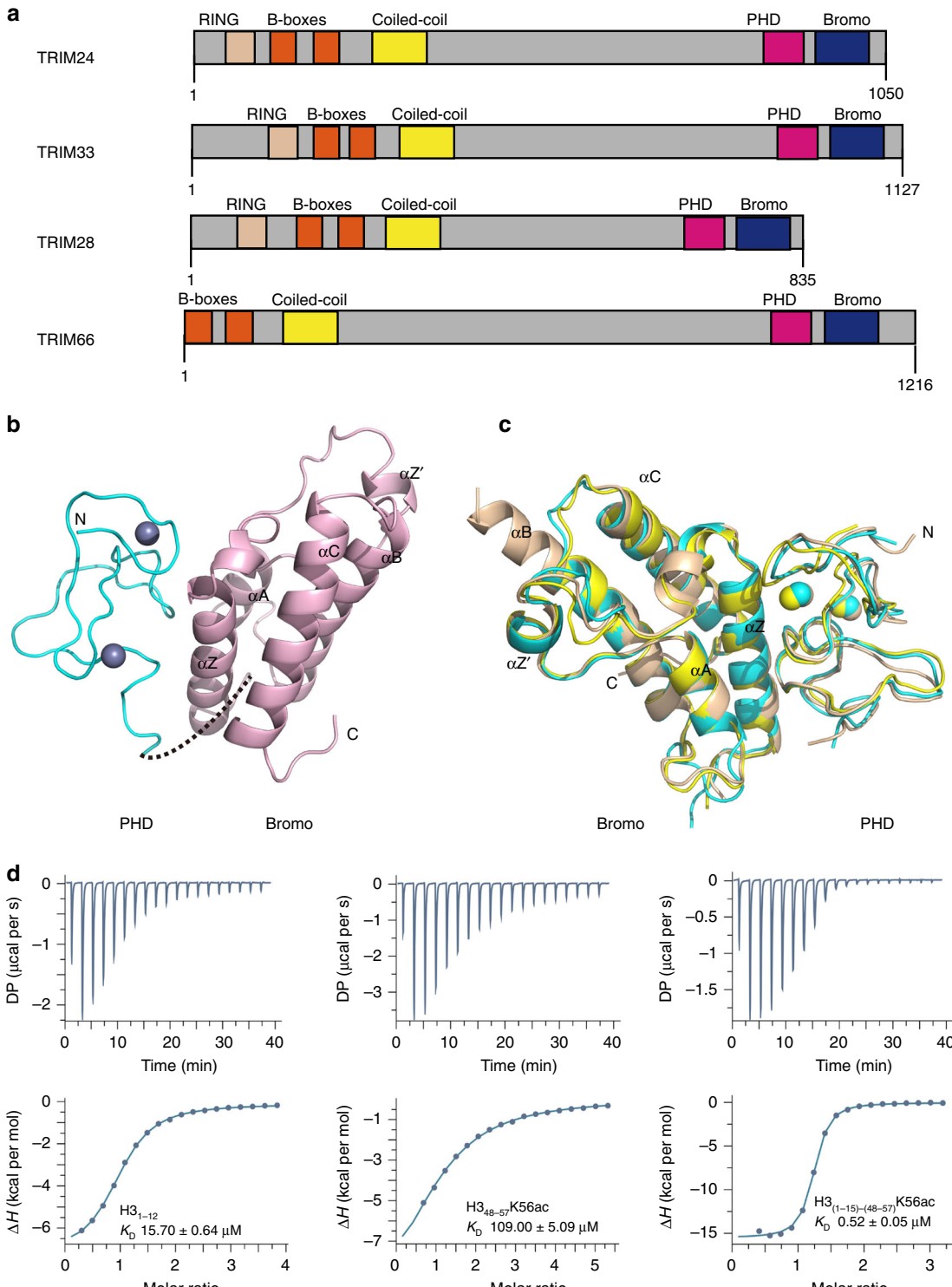

**Fig. 1** The TRIM66 PHD-Bromodomain interacts with the unmodified H3 N-terminal and H3K56ac. **a** The domain architecture of human TRIM24, TRIM28, TRIM33, and TRIM66. B-boxes, B-boxes domain; Bromo, bromodomain; Coiled-coil, coiled-coil region; PHD, plant homeodomain; RING, RING-finger domain. **b** Crystal structure of the TRIM66 PHD-Bromodomain in its free state. The linker (dash line) is invisible in the structure. The PHD finger (in aquamarine) and Bromodomain (in light pink) are graphically shown. Two zinc ions of the TRIM66 PHD are shown as silver spheres. **c** Superposition of the crystal structures of the PHD-Bromo tandem domains of TRIM24, TRIM33, and TRIM66. TRIM24 (cyan; PDB code: 3O33; 2.0 Å), TRIM33 (wheat; PDB code; 3U5M; 3.1 Å), and TRIM66 (yellow; this study; 2.1 Å) in their free state. **d** ITC titration: titrating TRIM66-WT$_{965-1160}$ with various histone H3 peptides

## Table 1 Data collection and refinement statistics

| | TRIM66-MUT$_{968-1160}$ | TRIM66-MUT$_{968-1160}$-H3$_{1-12}$ |
|---|---|---|
| Data collection | | |
| Space group | P6$_1$ | C2 |
| Cell dimensions | | |
| $a, b, c$ (Å) | 54.192, 54.192, 108.31 | 100.266, 63.465, 33.303 |
| $\alpha, \beta, \gamma$ (°) | 90, 90, 120 | 90, 101.674, 90 |
| Resolution (Å) | 23.47–2.101 | 30.12–1.787 |
| | (2.176–2.101)$^a$ | (1.851–1.787) |
| $R_{merge}$ | 12.8 (43.3) | 11.6 (64.0) |
| $I/\sigma I$ | 34.33 (6) | 22.41 (2.37) |
| Completeness (%) | 100 (100) | 99.5 (98.8) |
| Redundancy | 20.5 (20.9) | 4.2 (3.9) |
| Refinement | | |
| Resolution (Å) | 23.47–2.101 | 30.12–1.787 |
| No. reflections | 10,413 (1026) | 19,103 (1684) |
| $R_{work}/R_{free}$ | 20.72/24.74 | 17.52/21.71 |
| No. of atoms | | |
| Protein | 1269 | 1419 |
| Ligand/ion | 0/2 | 79/2 |
| Water | 56 | 124 |
| $B$-factors | | |
| Protein | 40.332 | 37.071 |
| Ligand/ion | 0/36.145 | 50.986/29.465 |
| Water | 41.426 | 43.763 |
| R.m.s. deviations | | |
| Bond lengths (Å) | 0.008 | 0.006 |
| Bond angles (°) | 0.9 | 0.83 |

The data set is from a single crystal
$^a$Values in parentheses are for highest-resolution shell

TRIM66-WT$_{965-1160}$ with these acetylated peptides, except for the H3$_{48-57}$K56ac peptide, which bound to TRIM66-WT$_{965-1160}$ with a $K_D$ of 109 μM (Fig. 1d and Supplementary Fig. 3a). We further found the acetylated state of K56 was necessary for this binding (Supplementary Fig. 3b).

As the PHD-Bromodomain of TRIM66 bound unmodified H3 tails and H3K56ac peptide, respectively, we hypothesized whether the unified PHD-Bromodomain bound these two peptides in a combinational manner. Therefore, we attempted to titrate a combinational peptide, which links H3$_{1-15}$ with H3$_{48-57}$K56ac (H3$_{(1-15)-(48-57)}$K56ac), to TRIM66-WT$_{965-1160}$ using ITC assays. As expected, the $K_D$ was 0.52 μM (Fig. 1d), 30-fold higher than the affinity for the H3$_{1-12}$ peptide and 210-fold higher than that for the H3$_{48-57}$K56ac peptide. Collectively, the PHD-Bromodomain binds to the unmodified H3 N-terminal and H3K56ac in a combinational manner.

**The H3R2me0K4me0 dual recognition module of TRIM66 PHD.** To identify the structural details of the TRIM66 PHD-Bromodomain binding the unmodified H3 tail, we also determined the 1.8Å crystal structure of the TRIM66-MUT$_{968-1160}$ in complex with the H3$_{1-12}$ peptide (Table 1). In this structure, the residues R2 to Q5 of the H3$_{1-12}$ peptide formed an anti-parallel β-sheet with the PHD finger of TRIM66. The residues T6 and K9 of the peptide contacted the residues D971, L977, N978, and G980 of the PHD finger (Fig. 2a, b). Both side chains of the R2 and K4 were docked on the negatively charged groove of the PHD finger (Fig. 2c). The side chain of R2 was hydrogen-bonded with D986, C985, E968, and N969, as well as a bridging water molecule (Fig. 2b). Moreover, we observed hydrogen bond interaction between the side chain of H3K4 and the side chain of D971 (Fig. 2b). The 2Fo–Fc electron density map for peptide showed that the electron density fit the model well (Supplementary

Fig. 4). These findings indicated that the D986 and D971 interacted with the R2 and K4 of the H3$_{1-12}$ peptide, playing important roles in the binding of PHD-Bromodomain for the unmodified H3$_{1-12}$.

The presence of a hydrogen bond network between the TRIM66 PHD domain and the H3R2 suggested the PHD domain may specifically recognize the H3R2 residue. We next titrated the TRIM66-WT$_{965-1160}$ with a series of R2-methylated H3 peptides through ITC assays. Intriguingly, the symmetrically dimethylated H3R2 (H3R2me2s) peptide and the asymmetrically dimethylated H3R2 (H3R2me2a) peptide demonstrated an obvious reduction in their binding affinities to TRIM66, whereas the monomethylated H3R2 (H3R2me1) bound to the TRIM66 PHD-Bromodomain with low affinity (Table 2, Supplementary Fig. 5a, and Supplementary Table 2). This finding was consistent with the structural observation that each NηH atom of the R2 formed a hydrogen bond with residues in protein (Fig. 2d). Further comparison of the H3R2 recognition mode between the TRIM66 PHD domain and PHD domains reported as H3R2me0 readers[43–45] revealed that the recognition modes of unmodified H3R2 were similar among these proteins (Supplementary Fig. 6). These results suggested the PHD domain of TRIM66 is a reader of unmodified H3R2.

We subsequently compared the complex structure of TRIM24, TRIM33, and TRIM66 with H3 peptide. Surprisingly, none or limited hydrogen bonding was observed between the H3R2 residue and the TRIM24 or TRIM33 PHD domain[5,6] (Fig. 2d). ITC assays using the TRIM24/TRIM33 PHD-Bromodomain with differently methylated H3R2 peptides revealed that both PHD-Bromodomains were insensitive to either H3R2me1 or H3R2me2a. However, they were sensitive to H3R2me2s (Table 2, Supplementary Fig. 5b, c, and Supplementary Table 2). Collectively, these results showed that the TRIM66 PHD exhibits a unique recognition of H3R2 among TRIM24, TRIM33, and TRIM66.

Hydrogen bonds or salt bridges were observed between the side chain of H3K4 and the conserved sequence Asn-Glu-Asp of PHDs in TRIM24, TRIM33, and TRIM66[5,6] (Fig. 2b). Therefore, we hypothesized that the PHD domain of TRIM66 may also specifically recognize H3K4me0. We titrated the TRIM66-WT$_{965-1160}$ with monomethylated, dimethylated, or trimethylated H3K4 through ITC assays and we observed a great reduction in binding affinity (Table 2, Supplementary Fig. 5d, and Supplementary Table 2). In short, the PHD-Bromodomain of TRIM66 specifically recognizes both unmodified H3R2 and H3K4, whereas TRIM24 and TRIM33 only recognize unmodified H3K4.

**H3K56ac peptide binds the ZA and BC loops of TRIM66 Bromo.** To investigate the interaction between the H3K56ac peptide and the TRIM66 PHD-Bromodomain, we performed a nuclear magnetic resonance (NMR) titration experiment. We created a mutant protein (termed TRIM66-GSGS$_{968-1160}$; Supplementary Fig. 1e, Supplementary Fig. 7a, b, Supplementary Table 1, and Supplementary Note 2). Although we observed a modest binding affinity ($K_D = 109.00$ μM) in the interaction between the H3K56ac peptide and TRIM66 PHD-Bromodomain (Fig. 1d), we observed obvious NMR chemical shift perturbations (Fig. 2e and Supplementary Fig. 7c) and several peaks even disappeared due to the NMR intermediate exchange effect. To determine the residue types of these perturbed peaks, we successfully assigned ca. 91.4% of all residues (proline excluded) using NMR spectroscopy (Supplementary Fig. 7d and Supplementary Note 3).

Subsequently, we mapped the residues with strong perturbations on the crystal structure of the TRIM66 PHD-Bromodomain, as shown in Fig. 2f. The results showed that the H3K56ac peptide

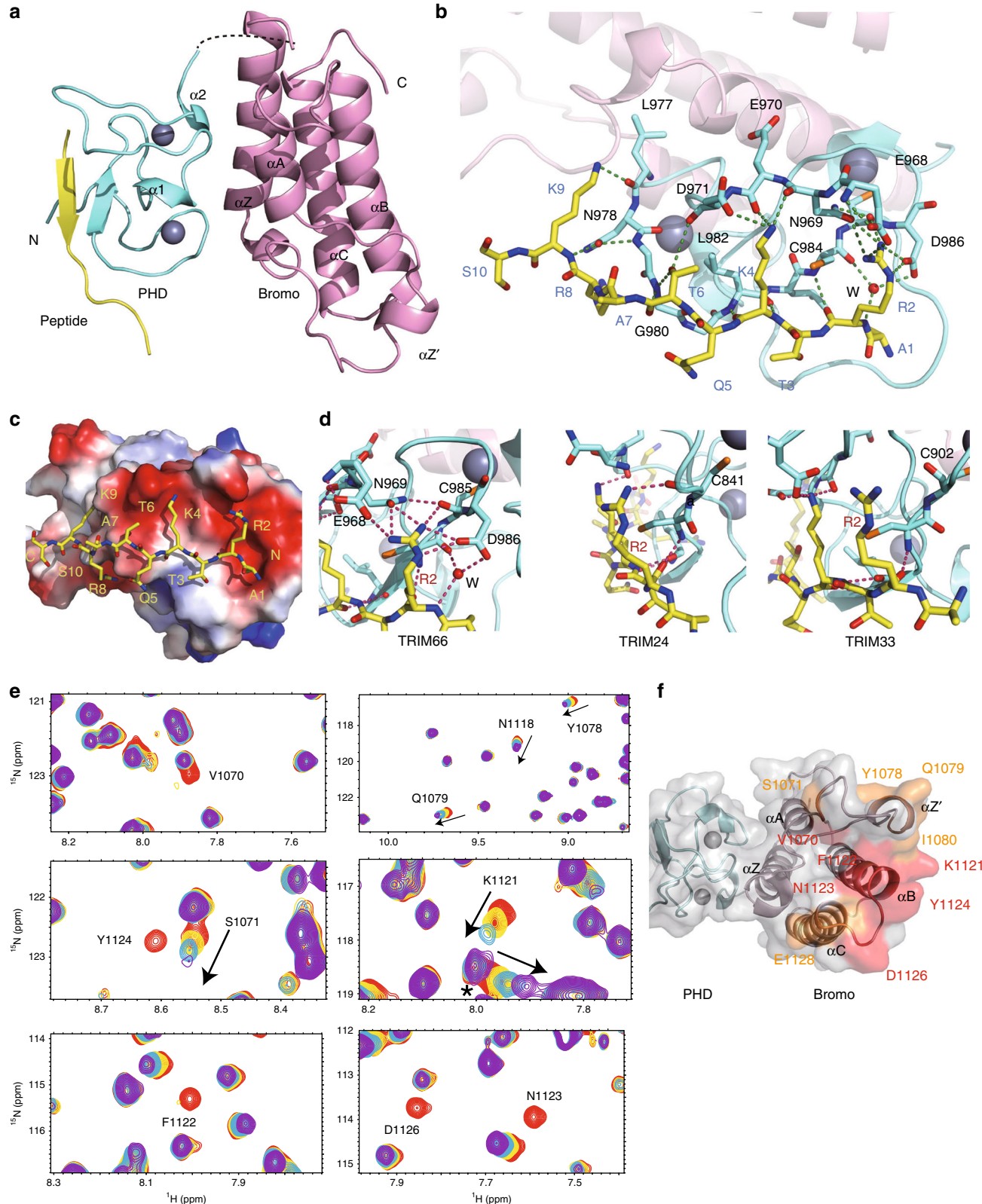

interacted with the ZA and BC loops of the TRIM66 Bromodomain. Residues F1122, N1123, Y1124, and D1126, which exhibited the most obvious perturbations, were all located on the BC loop, whereas V1070, S1071, Y1078, Q1079, I1080, and I1081 were located on the ZA loop. The ZA and BC loops are also critical for binding acetylated peptides in the Bromodomain of P/CAF, Gcn5p, TRIM24, CBP, and other Bromo-containing proteins[6,10,46,47]. One asparagine on the BC loop (N1123 in TRIM66) was conserved and played critical roles in recognizing acetyl-lysine. The amide nitrogen of the conserved asparagine usually formed hydrogen bond with the oxygen of the acetyl group of acetyl lysine. Our results suggested that the H3K56ac peptide interacts with the ZA and BC loops of the TRIM66 Bromodomain.

**Fig. 2** The structural details of the TRIM66 PHD-Bromodomain binding the H3 N-terminal and H3K56ac peptides. **a** Crystal structure of the TRIM66 PHD-Bromodomain in the complex state with H3 N-terminal tail. The H3 N-terminal peptide (in yellow) interacts with the PHD finger domain (in aquamarine). Bromodomain is shown in light pink. The zinc ions are shown as silver spheres. The linker (dash line) is invisible in the structure. **b** Detailed interactions between the $H3_{1-10}$ peptide (in yellow) and the PHD finger of TRIM66 in the complex structure of the TRIM66 PHD-Bromodomain binding the H3 N-terminal tail. The hydrogen bonds are shown in forest green dashed lines. In **b–d**, several residues in the interaction between the TRIM66 PHD finger and N-terminal H3 tail are labeled and shown in sticks with red oxygen atoms, blue nitrogen atoms, and orange sulfur atoms. A water molecule mediating the H3-TRIM66 PHD interaction is shown as a small red sphere. **c** Electrostatic (protein) and stick (peptide) representation of the crystal structure of $H3_{1-10}$ peptide binding to the TRIM66 PHD-Bromodomain. **d** Detailed interactions between the H3R2 residue and the PHDs from TRIM24, TRIM33, and TRIM66. The hydrogen bonds are shown in warm pink dashed lines. TRIM24, PDB code: 3O37; TRIM33, PDB code: 3U5O; TRIM66, this study. **e** Close-up views of the NMR chemical shift perturbations of the $^{15}N$-labeled TRIM66 PHD-Bromodomain upon the titration of the H3K56ac peptides. The molar ratio of peptide/protein was 0:1 (red), 0.7:1 (yellow), 1.4:1 (blue), and 2.8:1 (purple). The perturbation direction is marked as the arrow and the corresponding residues are also labeled. The unassigned residues are marked as a star. **f** The perturbed residues were mapped and labeled on the structure of the TRIM66 PHD-Bromodomain (gray surface). Residues F1122, N1123, Y1124, and D1126 are shown in red, whereas other perturbed residues are shown in yellow

| Table 2 Binding affinities of different histone peptides to the PHD-Bromodomain of TRIM proteins by ITC | | | |
|---|---|---|---|
| **Histone peptides** | $K_D$ **(μM)** | | |
| | **TRIM66** | **TRIM24** | **TRIM33** |
| $unH3_{1-12}$ | 15.70 ± 0.64 | 51.40 ± 3.49 | 3.39 ± 0.08 |
| H3R2me1 | 72.40 ± 3.27 | 72.70 ± 2.23 | 5.10 ± 0.15 |
| H3R2me2a | 110.00 ± 11.90 | 76.80 ± 1.20 | 6.23 ± 0.23 |
| H3R2me2s | aWeak binding | Weak binding | Weak binding |
| H3K4me1 | 42.80 ± 2.45 | – | – |
| H3K4me2 | Weak binding | – | – |
| H3K4me3 | Weak binding | – | – |

$K_D$ values were calculated from single measurement and errors were estimated from fitting curve by MicroCal PEAQ-ITC analysis software
aITC curves cannot be fitted reliably

**Loss of TRIM66 leads to DNA damage and genomic instability**. TRIM66 serves as a new chromatin reader for H3K56ac, which plays a vital role in DDR and genomic integrity from yeast to mammals, and is engaged in the pluripotency regulation of ESCs[30,31,35–37]. Therefore, we investigated the role of TRIM66 in ESCs owing to its decreasing expression during the embryoid body (EB) differentiation (Supplementary Fig. 8a, b). We then established two Trim66-knockout (KO) ESC lines (termed as TRIM66 KO-1 and KO-2) via CRISPR/Cas9 (Supplementary Fig. 8c, d). Compared with wild-type (WT) ESCs, TRIM66-KO ESCs did not display overt changeable morphology, expression of pluripotency markers, alkaline phosphatase activity, and EB-forming ability (Supplementary Fig. 8e–h). These findings suggested that loss of TRIM66 does not impair the self-renewal of ESCs.

However, a robust increase in the level of γ-H2AX, an indicator of DSB damage[48,49], and formation of foci were detected in TRIM66-KO ESCs (Fig. 3a, b). Furthermore, we generated a doxycycline (Dox)-inducible ESC line (shTrim66 tet-on). Our results showed that the level of γ-H2AX was continuously elevated in the shTrim66 tet-on ESCs treated with Dox for 96 h. Following the Dox withdrawal at 48 h, the accumulated level of γ-H2AX was gradually reduced along with the recovery of TRIM66 expression (Fig. 3c). We further performed the comet assay to measure the extent of DNA damage on a single-cell basis. This experiment showed more severe DNA damage in TRIM66-KO ESCs (Fig. 3d, e). Furthermore, karyotype analysis of chromosome metaphase spreads revealed that the TRIM66-KO ESCs exhibited severe chromosomal abnormalities, including a higher rate of chromosomal breakage and chromosome ends fusion (Fig. 3f, g). Meanwhile, the expression of a series of genes (Cdkn1a, Gadd45a, and p53)[50–52] related to DNA damage

response was increased in TRIM66-KO ESCs (Supplementary Fig. 10a). In addition, these phenotypes of increased DNA damage and chromosomal abnormalities were reproducibly observed following the knockdown of Trim66 by short hairpin RNAs (Supplementary Fig. 9). Collectively, these results indicated that TRIM66 is critical for DDR and maintaining genomic stability in ESCs.

In view of severe DNA damage caused by TRIM66 depletion in ESCs, it would be vital to figure out whether TRIM66 was required for ESC to survive DNA damage. Consequently, we treated the TRIM66-KO ESCs with several DNA-damaging agents including etoposide (EPI), ionizing radiation (IR), or bleomycin, and our results showed that TRIM66 depletion significantly reduced the survival rates of ESCs (Supplementary Fig. 10b). TRIM66-KO ESCs also inhibited the cell growth of ESCs with EPI treatment (Supplementary Fig. 10c, d). Furthermore, we performed competitive survival experiment through mixing the equal amount of TRIM66-KO ESCs and green fluorescent protein (GFP)-labeled WT ESCs (WT-GFP), then co-cultured for 5 days under EPI treatment. Our results showed that the ratio of TRIM66-KO ESCs was gradually decreased, suggesting that TRIM66 was indeed required for ESC to survive DNA damage (Supplementary Fig. 10e).

**TRIM66 maintains genomic integrity through PHD-Bromodomain**. We further investigated the role of the TRIM66 PHD-Bromodomain in safeguarding the genome. We first introduced a series of single amino acid mutations to the PHD domain at interaction interface and then performed corresponding ITC assays. The mutant E968A and N969A showed slightly lower binding affinity to the $H3_{1-12}$ peptide compared with the TRIM66-$WT_{965-1160}$, whereas the mutant D986A nearly abolished its ability of binding. Subsequent ITC assays revealed a weak binding of the mutant D971A for the H3 peptide (Fig. 4a). We also mutated the conserved asparagine residue N1123 in TRIM66, which also demonstrated obvious perturbations in titration. ITC assays showed that the N1123A mutant abolished the ability to bind the H3K56ac peptide (Fig. 4b), suggesting the critical role of N1123 in binding the H3K56ac peptide.

As mentioned earlier, the TRIM66 PHD-Bromodomain bound the $H3_{(1-15)-(48-57)}$K56ac peptide with a higher affinity than that for the individual peptides (Fig. 1d). Considering the structural difference of peptide, histone, and nucleosome, we also tried to analyze the binding ability of the TRIM66 PHD-Bromodomain for histones and the nucleosome. We observed the interaction of TRIM66 PHD-Bromodomain with histones using GST-pulldown assays (Fig. 4c). We added histone chaperone $ASF1a_{1-156}$ to stabilize the H3K56ac-H4 complex (Fig. 4d)[53] and prevent the precipitation of the complex during the experiment. Use of the mutant D986A/N1123A, N1123A, and D986A instead of the

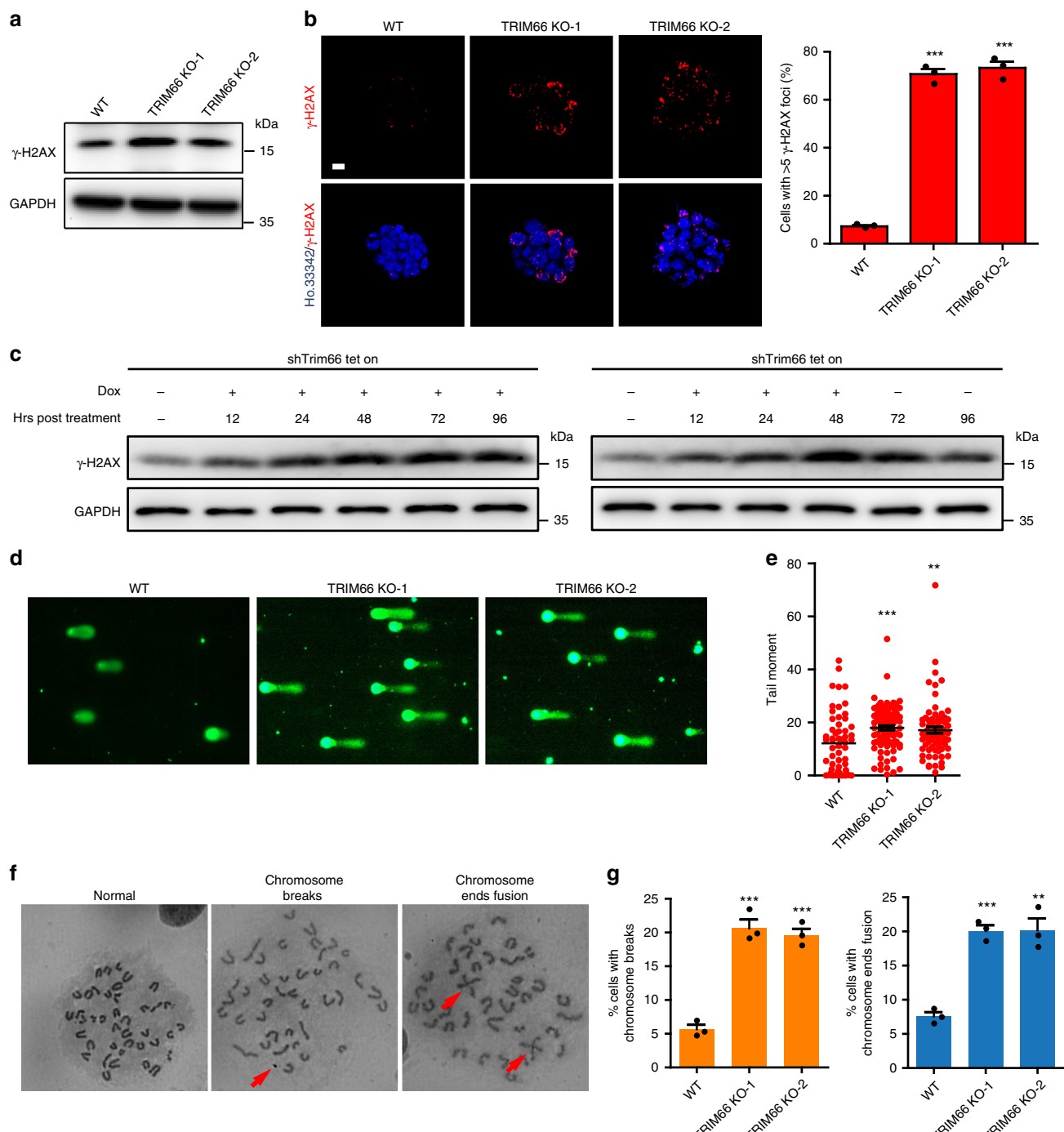

**Fig. 3** Loss of TRIM66 leads to severe DNA damage and genomic instability in ESCs. **a** Immunoblot analysis showing the levels of γ-H2AX in WT and TRIM66-KO ESCs. **b** The formation of γ-H2AX foci in WT and TRIM66-KO ESCs. Representative immunofluorescence images (left) and the quantification (right) are shown. Scale bar, 10 μm. More than 200 cells are examined in each sample. **c** The levels of γ-H2AX are detected by immunoblot analysis in shTrim66 tet-on ESCs. ESCs are persistently treated with doxorubicin (Dox, 1 μg/ml) for 96 h (left) or 48 h followed by withdrawal of Dox (right). **d** Representative comet assay images of WT and TRIM66-KO ESCs. **e** DNA integrity assessment of WT and TRIM66-KO ESCs by the comet assay in **d**. More than 100 cells are examined in each sample. **f** Representative gray-scale images of chromosome metaphase spreads of WT and TRIM66-KO ESCs. Arrowheads in red indicates chromosomal breakage or chromosome ends fusion. **g** Quantification of chromosomal breakage (left) or chromosome ends fusion (right) in WT and TRIM66-KO ESCs in **f**. More than 200 cells are examined in each sample. Data are presented as the means ± SEM. Statistical significance is determined by two-tailed Student's $t$-test. $**p < 0.01$, $***p < 0.001$. Source data are provided as a Source Data file

TRIM66-WT$_{965-1160}$ protein showed that the two single point mutants bound H3K56ac-H4-ASF1a with lower binding ability compared with that observed for the WT (Fig. 4c). Notably, the double point mutant (D986A/N1123A) showed the lowest level of complex binding. These results suggested that both the PHD and

Bromodomain contributed to the binding of the H3K56ac-H4-ASF1a complex. Unexpectedly, we did not observe binding between the TRIM66 PHD-Bromodomain and nucleosome using ITC, GST pulldown, or electrophoretic mobility shift assay. Therefore, the TRIM66 PHD-Bromodomain could bind the

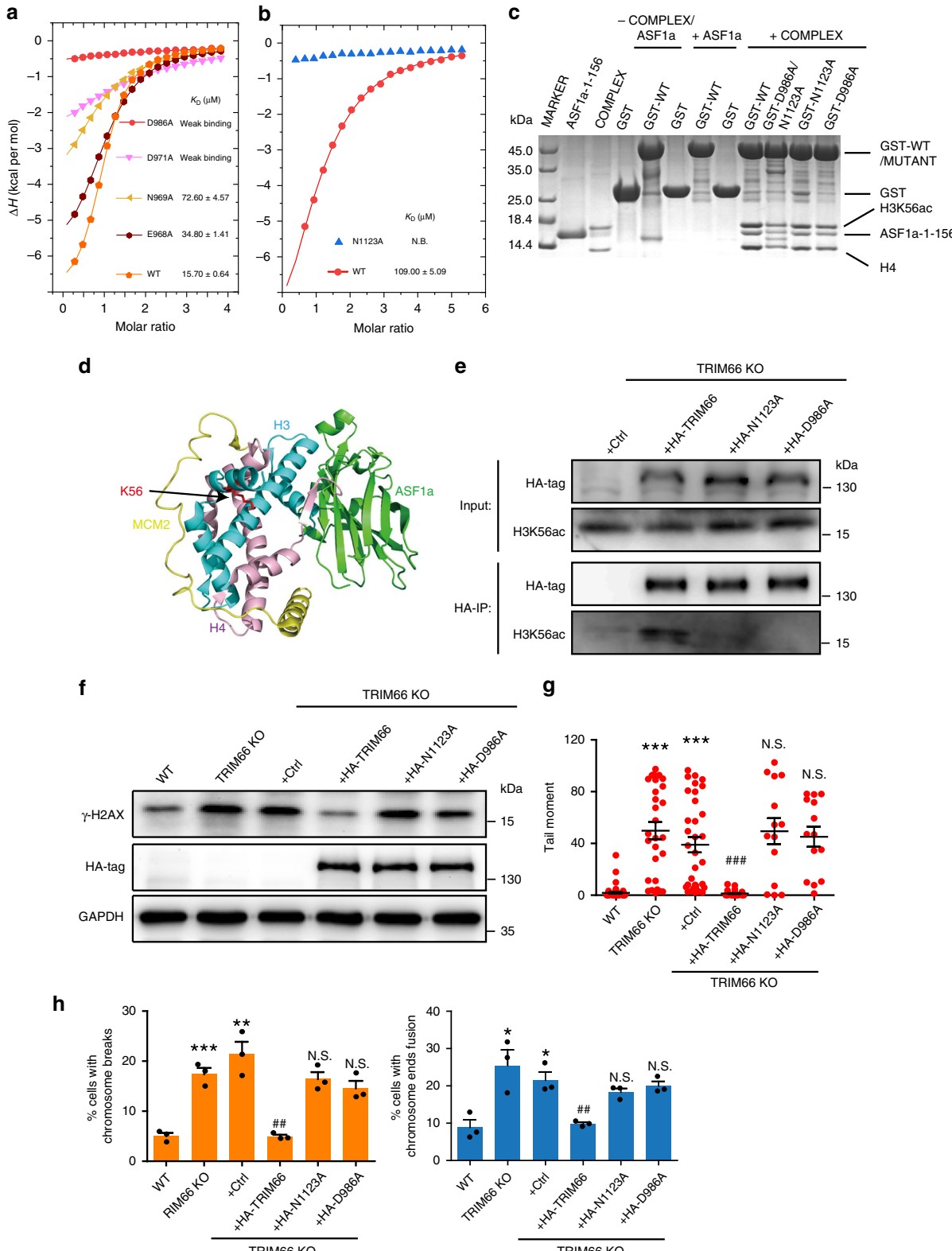

ligated H3R2me0K4me0-H3K56ac peptide and the H3K56ac-H4 dimer in a combinational manner.

Further, we reconstituted HA-tagged WT or mutant TRIM66 (N1123A and D986A) in TRIM66-KO ESCs and verified the interaction between TRIM66 and H3K56ac in ESCs by co-immunoprecipitation (Co-IP). Our results revealed that WT

TRIM66 could bind to H3K56ac, whereas TRIM66 mutants apparently lost their ability to bind to H3K56ac (Fig. 4e). Meanwhile, we analyzed the colocalization of H3K56ac and TRIM66-WT or mutants in ESCs. Our results found that, unlike the N1123A and D986A mutants, the WT TRIM66 displayed fine colocalization with H3K56ac in the nucleus of ESCs by

**Fig. 4** TRIM66 maintains genomic integrity through its PHD-Bromodomain. **a** ITC titration: titrating TRIM66-WT$_{965-1160}$ and mutant proteins with unmodified H3$_{1-12}$ peptides. **b** ITC titration: titrating TRIM66-WT$_{965-1160}$ proteins and its mutant N1123A with H3K56ac peptides. **c** The GST pulldown assays of the GST-TRIM66-WT$_{965-1160}$ and its mutants against the H3K56ac-H4-ASF1a complex. **d** The overall structure of the H3-H4-ASF1a-MCM2 quaternary complex (PDB code: 5C3I). The molar ratio of H3-H4/ASF1a/MCM2 is 1:1:1. The lysine 56 of H3 (shown in stick and red) is exposed to the solution. H3 (in aquamarine), H4 (in warm pink), MCM2 (in yellow), and ASF1a (in green) are labeled. **e** Anti-HA immunoprecipitates are prepared from ESC-overexpressed HA-tagged WT TRIM66 and point mutations followed by immunoblot analysis. **f** Immunoblot analysis of the levels of γ-H2AX in WT, TRIM66-KO ESCs, and the TRIM66-KO ESC-overexpressed control plasmid (Ctrl), HA-tagged WT TRIM66, the N1123A, or D986A mutant of TRIM66. **g** DNA integrity assessment of ESCs as described in **f** by the comet assay. More than 100 cells are examined in each sample. **h** Quantification of chromosomal breakage (left) or chromosome ends fusion (right) in WT, TRIM66-KO ESCs, and the rescued TRIM66-KO ESCs. More than 200 cells are examined in each sample. Statistical significance is determined by two-tailed Student's $t$-test. * represents a difference from WT ESCs, # indicates a difference from TRIM66-KO + Ctrl. Data are presented as the means ± SEM. *$p < 0.05$, **##$p < 0.01$, ***###$p < 0.001$; N.S., no significance. Source data are provided as a Source Data file

quantifying the fluorescence intensity[54] (Supplementary Fig. 11). Furthermore, our results showed that the WT TRIM66, but not the mutants, restored the elevated level of γ-H2AX in TRIM66-KO ESCs (Fig. 4f). Correspondingly, only the induction of WT TRIM66 restored the excessive DNA damage (Fig. 4g). Similarly, these two mutants failed to reduce the rates of chromosomal breakage and chromosome ends fusion in TRIM66-KO ESCs (Fig. 4h). Collectively, our results demonstrated that TRIM66 is involved in DDR and maintenance of genomic stability through its recognition of the H3R2me0K4me0-H3K56ac via the PHD-Bromodomain.

**TRIM66 KO causes the H3K56ac retention following DNA damage**. We subsequently investigated the regulation of DDR by TRIM66 through recognition of H3K56ac in ESCs. Initially, we measured the variation of H3K56ac level in the shTrim66 tet-on ESCs persistently treated with Dox for 96 h and found that the expression was continuously upregulated (Fig. 5a). Intriguingly, the increased level of H3K56ac was slightly reduced after Dox withdrawal at 48 h, suggesting that the level of H3K56ac was negatively correlated with TRIM66 in ESCs (Fig. 5a). H3K56ac is actively deacetylated at DNA damage sites at the early phase of DNA repair signaling[32,35]. Immunofluorescent staining revealed that the level of H3K56ac was reduced after treatment with EPI for 1 h in WT ESCs, whereas the decline rate of H3K56ac level was drastically retarded in TRIM66-KO ESCs (Fig. 5b). TRIM66 was significantly enriched to chromatin after stimulation with EPI, in parallel with the decreasing level of H3K56ac in the chromatin fraction (Fig. 5c). Further, we found that the increased TRIM66 foci, induced by DNA-damaging agents (EPI or IR), were colocalized with γ-H2AX foci (Supplementary Fig. 12a, b). Notably, our results showed that WT ESCs displayed a proper competence in DDR, whereas TRIM66-KO ESCs exhibited an impairment in DDR, suggesting that TRIM66 was critical for DNA damage recovery of ESCs (Supplementary Fig. 12c). Together, these results suggested that TRIM66 is recruited to chromatin and deacetylated H3K56ac following DNA damage.

Then, we detected the level of H3K56ac was clearly reduced at DNA damage sites (γ-H2AX foci), whereas it was mostly present at the γ-H2AX foci in TRIM66-KO ESCs analyzed by quantifying the fluorescence intensity (Fig. 5d, e). These data suggested that the absence of TRIM66 resulted in invalidated deacetylation of H3K56ac at DNA damage sites, highlighting the critical role of TRIM66 in facilitating DDR. In addition, the DDR response components, p-ATM, and the DNA repair-related factors including Rad51, CtIP, and p-DNA-PKcs[55–60], were recruited and colocalized with the γ-H2AX foci following EPI treatment in WT ESCs. In contrast, the colocalization of Rad51, CtIP, and p-DNA-PKcs, not p-ATM, with the γ-H2AX foci was completely abolished in TRIM66-KO ESCs (Fig. 5f, g and Supplementary Fig. 12d–g).

To determine the dependence of H3K56ac regulation by TRIM66 on its recognition, we detected the H3K56ac level after overexpressing WT and point mutant TRIM66 in TRIM66-KO ESCs. Our data showed that only the overexpression of WT TRIM66 in TRIM66-KO ESCs restored the level of H3K56ac (Fig. 5h). Furthermore, these recognition mutants of TRIM66 failed to rescue the high incidence of H3K56ac retention with γ-H2AX foci and the decreased colocalization of DNA repair-related factors with γ-H2AX foci in TRIM66-KO ESCs (Fig. 5i, j). Overall, these data revealed that TRIM66 reads and reduces the level of H3K56ac, facilitating the recruitment of the downstream DNA repair machinery at DNA damage sites.

**TRIM66 recruits Sirt6 to safeguard genomic stability**. We speculated that TRIM66 regulated the deacetylation of H3K56ac at DNA damage sites through recruiting Sirt1 or Sirt6, which are responsible for deacetylating H3K56ac in mammals[33,36,61]. We performed Co-IP between TRIM66 and Sirt1 or Sirt6 in ESCs. The results showed that the endogenous TRIM66 protein could associate with Sirt6 (Fig. 6a), which was further confirmed through exogenous Co-IP assay (Fig. 6b). More importantly, the accumulated recruitment of Sirt6 to chromatin following EPI treatment, consistent with previous findings[35], was significantly inhibited in TRIM66-KO ESCs (Fig. 6c). These results indicated that TRIM66 is responsible for the recruitment of Sirt6 on chromatin in response to DNA damage.

Otherwise, we excluded the possibility that the point mutants could affect the immunoprecipitation with Sirt6 (Supplementary Fig. 13). Further, we generated 293T cells efficiently expressing Sirt6 (Flag-tagged) and truncations of TRIM66 (HA-tagged F1$_{1-240}$, F2$_{241-916}$, and F3$_{916-1216}$) (Fig. 6d). Our results showed that Sirt6 could be immunoprecipitated by the full-length TRIM66 and truncations including the F1 + F2, F2 + F3, and F2 (Fig. 6d, e). Thus, we deduced that the F2 region is required for the association of TRIM66 with Sirt6.

Subsequently, we generated TRIM66-KO ESC lines over-expressing these truncations. Our results showed that the overall levels and foci formation of γ-H2AX were restored by over-expression of either the full length of TRIM66 or the F2 + F3 truncation, which contained both the Sirt6-associated region and histone PTM recognition region (Fig. 6f, g). Similarly, only the full length of TRIM66 and the F2 + F3 truncation could recover the level of H3K56ac in TRIM66-KO ESCs (Fig. 6f, h). Correspondingly, the increasing intensity of DNA damage and percentage of chromosomal breakage and chromosome ends fusion following Trim66 depletion were rescued by the exogenous expression of WT and the F2 + F3 truncation of TRIM66 (Fig. 6i, j). Collectively, our findings indicated that both PTM recognition and Sirt6 recruitment by TRIM66 are indispensable for the regulation of DDR and maintenance of genomic integrity. Therefore, we propose a model in which TRIM66 reads the

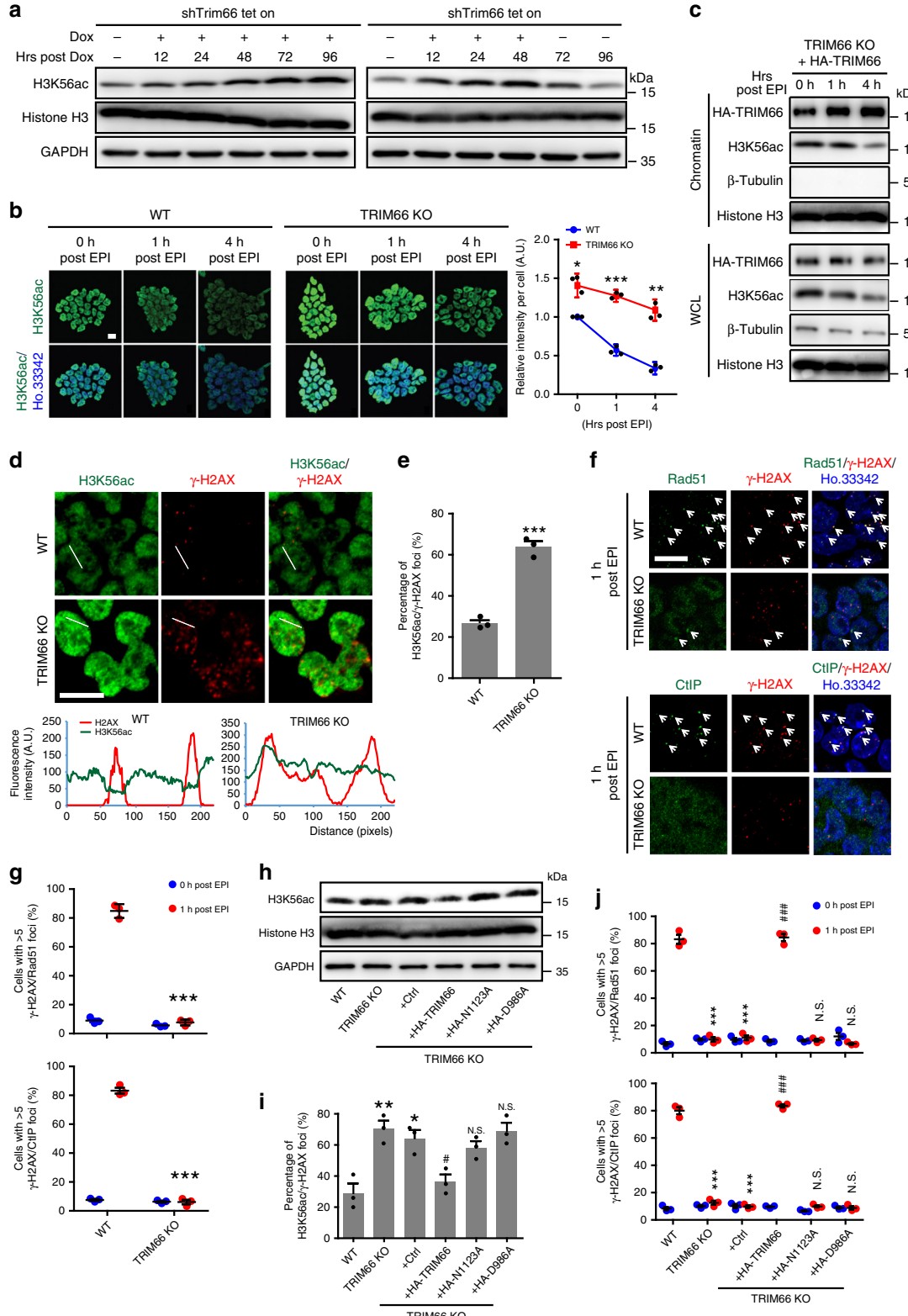

specific H3R2me0K4me0-H3K56ac modifications and recruits Sirt6 to chromatin for the deacetylation of H3K56ac at DNA damage sites. This process facilitates the subsequent recruitment of DNA repair proteins in response to DNA damage.

**Loss of TRIM66 results in genomic instability in vivo.** Considering the importance of TRIM66 in the maintenance of

genomic stability, further investigation of the physiological role of TRIM66 in regulating murine development is necessary. Notably, the birth rate of single KO (Trim66$^{+/-}$) mice and double KO mice (Trim66$^{-/-}$) was in accordance with Mendel's law of separation after heterozygote crossing (Fig. 7a). However, we found that the body weight of the Trim66-deficient offspring, aged 2 or 3 weeks, was significantly decreased compared with that

**Fig. 5** Depletion of Trim66 results in the retention of the level of H3K56ac following DNA damage. **a** Immunoblot analysis of H3K56ac in shTrim66 tet-on ESCs as described in Fig. 3c. **b** Representative immunofluorescence images (left) and quantification (right) of H3K56ac in WT and TRIM66-KO ESCs at different time points after treatment with etoposide (EPI, 10 μM for 30 min). Scale bar, 10 μm. More than 100 cells are examined in each sample. **c** Immunoblot analysis of cell fractionation in TRIM66-KO ESCs overexpressed with HA-tagged WT TRIM66 at different time points after treatment with EPI. **d** Representative immunofluorescence images of H3K56ac and γ-H2AX foci in WT and TRIM66-KO ESCs, and the intensity profile of both H3K56ac and γ-H2AX across the white line as shown in the images. Scale bar, 10 μm. **e** Quantification of the colocalized H3K56ac and γ-H2AX foci in WT and TRIM66-KO ESCs. More than 100 cells are examined in each sample. **f, g** Representative immunofluorescence images and quantification of colocalized Rad51/CtIP and γ-H2AX foci in WT and TRIM66-KO ESCs at 1 h after treatment with EPI. Scale bar, 10 μm. Arrowheads indicates colocalized foci. More than 100 cells are examined in each sample. **h** Immunoblot analysis of the H3K56ac in WT, TRIM66-KO ESCs, and the TRIM66-KO ESCs overexpressed control plasmid, HA-tagged WT TRIM66, the N1123A or D986A mutant of TRIM66. **i** Quantification of colocalized H3K56ac and γ-H2AX foci in ESCs described in **h**. More than 200 cells are examined in each sample. **j** Quantification of the colocalized Rad51/CtIP and γ-H2AX foci in ESCs described in **h** at 1 h after treatment with EPI. More than 100 cells are examined in each sample. *represents a difference from WT ESCs, #indicates a difference from TRIM66-KO + Ctrl. Data are presented as the means ± SEM. Statistical significance is determined by two-tailed Student's *t*-test. *#*p* < 0.05, **p* < 0.01, ***###*p* < 0.001, N.S., no significance. Source data are provided as a Source Data file

reported in Trim66[+/+] offspring (Fig. 7b). These results suggested that the depletion of Trim66 may delay whole-body development.

To further investigate the effect of TRIM66 on the genomic stability in early embryos, we isolated Trim66[+/+] blastocysts encompassing the inner cell mass from the pregnant mice at E3.5. In addition, Trim66-deficient blastocysts were isolated from the female Trim66[−/−] mice crossed by male Trim66[+/−] mice. Remarkably, we observed significant increased levels of H3K56ac and γ-H2AX in the Trim66-deficient blastocysts (Fig. 7c, d), consistent with the results observed in ESCs in vitro. Moreover, we derived the primary ESCs from the Trim66[+/+] and Trim66[−/−] embryos, and found that the levels of γ-H2AX and H3K56ac were significantly elevated in primary Trim66[−/−] ESCs (Fig. 7e–g). Meanwhile, the rates of chromosomal breakage and chromosome ends fusion in primary Trim66[−/−] ESCs were also increased compared with the Trim66[+/+] group (Fig. 7h). Collectively, we concluded that TRIM66 is required for the maintenance of genomic stability and early development in embryos.

## Discussion

The sequence of PHD is highly conserved in TRIM24, TRIM33, and TRIM66 (Supplementary Fig. 1a). We found that the PHD domain of TRIM66 recognized unmodified H3R2K4. Compared with other proteins of the C-VI TRIM family, we found that the side chain of the glutamic acid in TRIM24 and TRIM33—D986 in TRIM66—was excessively long to form hydrogen bonds with H3R2 (Fig. 2d). We supposed that the binding of D986 to H3R2me0 results in the formation of hydrogen bonds between C985 and N969, pulling the side chain of N969 close to H3R2me0. In addition, the N969 of TRIM66—highly conserved in TRIM24 and TRIM33—plays a role in recognizing H3R2me0. However, no contact has been reported between the corresponding asparagines and H3R2me0 in TRIM24 and TRIM33 (Fig. 2d)[5,6]. These results indicated that TRIM66 differs from other C-VI TRIM family members, specifically binding H3R2me0 via its PHD domain.

We verified that the TRIM66 Bromodomain recognized H3K56ac through its ZA and BC loops (Fig. 2e, f). The recognition of unmethylated H3R2K4 by the TRIM66 PHD domain might improve its binding affinity and specificity, and stabilize the interaction of the Bromodomain to histones containing the H3K56ac (Figs. 1d and 4c). More importantly, the recognition mutations on PHD or Bromodomain of TRIM66 failed to bind H3K56ac in ESCs (Fig. 4e and Supplementary Fig. 11). These findings suggested that the TRIM66 PHD-Bromodomain reads H3R2me0K4me0-H3K56ac modifications in a combinational manner.

ESCs are exposed to a greater risk of DNA damage due to their rapid proliferation and highly active transcription. It has been found that ESCs possess a unique and efficient system of DDR and maintenance of genomic stability, which lowers the mutation rate[52,62–64]. Our study found that TRIM66—expressed in ESCs—was significantly downregulated during ESC differentiation. The depletion or knockdown of Trim66 led to increased DNA damage and obvious karyotype abnormalities in ESCs. We also found that DNA damage was significantly increased in the Trim66-KO blastocysts and primary Trim66-KO ESCs, which may contribute to the slow development. Furthermore, our results also showed that TRIM66 could bind to the DNA damage sites and affect the localization of repair factors, whereas the point mutants were unable to restore the DNA damage, genomic instability, and repair factor mislocalization caused by Trim66 depletion. This result indicated that the regulation of DDR and maintenance of genomic stability depends on the recognition ability of TRIM66 for histone PTM. This is the first study to propose that TRIM66 may be an unrevealed regulator involved in DDR and maintenance of genomic stability in ESCs via the recognition of H3R2me0K4me0-K56ac modifications.

Previous studies indicated that H3K56ac is deacetylated at the damage sites, which is a prerequisite for the subsequent DDR[32–36]. It is established that histone deacetylases (e.g., sirtuins) are capable of deacetylating H3K56ac[34–36,65]. In particular, Sirt6 reportedly participates in DDR such as HR/NHEJ[35,66,67]. Our results showed that the H3K56ac level was negatively correlated with the expression of TRIM66 in ESCs and significantly increased in the Trim66-KO ESCs and blastocysts. In Trim66-depleted ESCs, the WT TRIM66, rather than TRIM66 mutants, rescued the upregulated levels of DNA damage and H3K56ac. Moreover, it effectively relieved the retention of H3K56ac and the blockage of the recruitment of downstream DNA repair factors at the damage sites. These results suggested that TRIM66 greatly participated in the H3K56ac deacetylation during DDR. More importantly, we found that TRIM66 may associate with Sirt6 and Trim66 depletion significantly attenuated the enrichment of Sirt6 on the chromatin in response to DNA damage. These findings indicated that TRIM66 may recruit Sirt6 to chromatin following DNA damage. Furthermore, our results showed that only the TRIM66 truncations containing both the Sirt6-interacting region and PHD-Bromodomain were able to rescue DNA damage augmentation, retention of H3K56ac at the damage sites, and genome instability caused by Trim66 depletion. Collectively, our studies confirmed that TRIM66 recognizes H3K56ac and recruits Sirt6 to deacetylate H3K56ac at the damage sites, thus initiating the DDR process (Supplementary Fig. 14).

We observed the binding of the PHD-Bromodomain of TRIM66 with H3K56ac peptide and histone but not nucleosome.

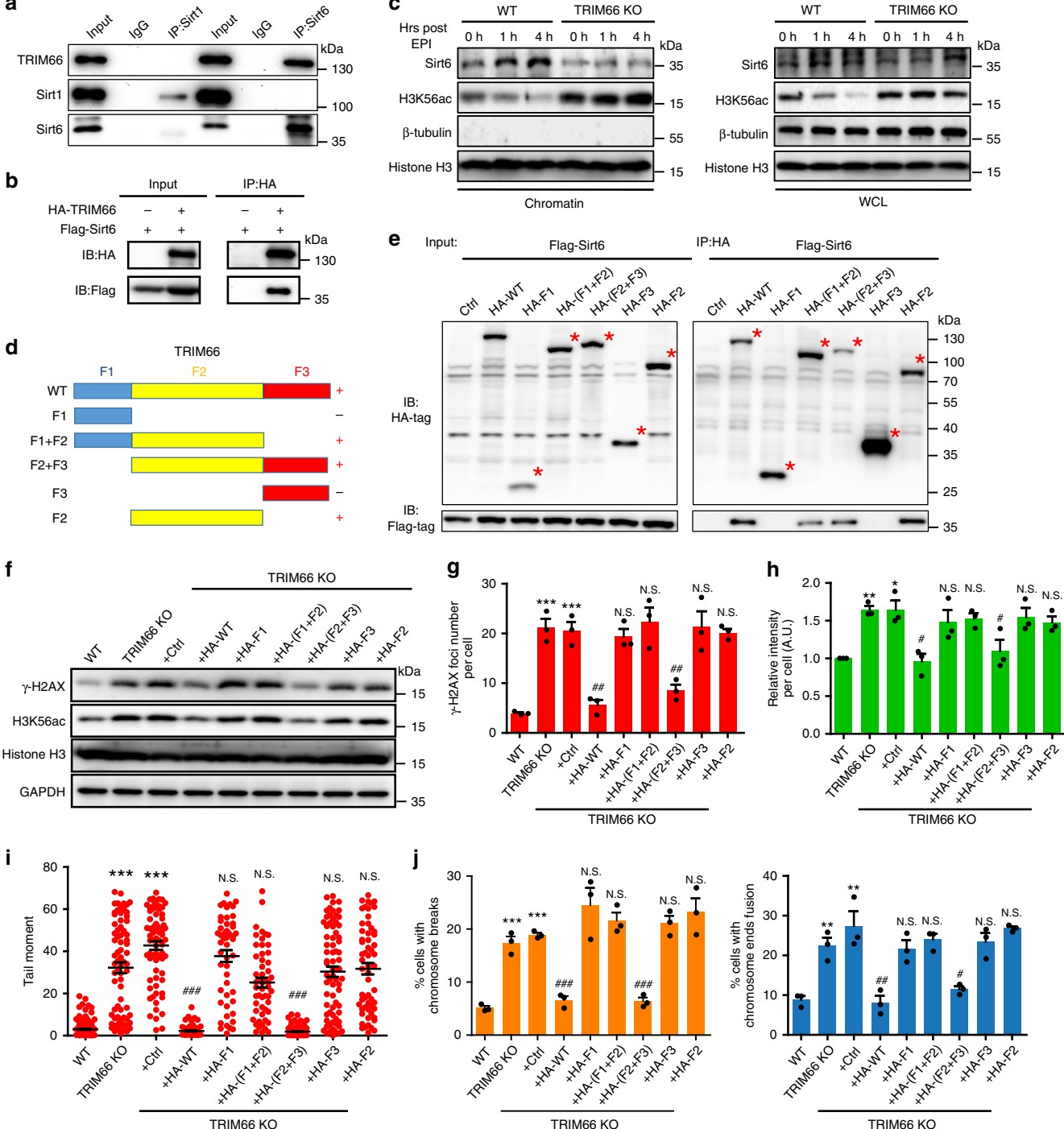

**Fig. 6** TRIM66 recruits Sirt6 to deacetylate H3K56ac and safeguard genomic stability in ESCs. **a**, Co-immunoprecipitation analysis between Sirt1/Sirt6 and TRIM66 in ESCs. **b** HA immunoprecipitation analysis in 293T cells showing the interaction between TRIM66 and Sirt6. **c** Immunoblot analysis of the cell fractionation in WT and TRIM66-KO ESCs at different time points after treatment with EPI. **d** Schematic representation of WT TRIM66 and the truncated mutant proteins. **e** Immunoblot analysis of WT TRIM66 and truncations associated with Sirt6, which is marked by red plus sign in **d**, whereas the truncations not interacting with Sirt6 are marked by the black minus sign. Anti-HA immunoprecipitates are prepared from 293T cells transfected with Flag-tagged Sirt6 and HA-tagged WT TRIM66, and truncations followed by immunoblot. **f** Immunoblot analysis of the levels of γ-H2AX and H3K56ac in WT, TRIM66-KO ESCs, and the TRIM66-KO ESC-overexpressed control plasmid, HA-tagged WT TRIM66, and the truncated mutants of TRIM66. **g** Quantification of the γ-H2AX foci formation in ESCs described in **f**. More than 200 cells are examined in each sample. **h** Quantification of H3K56ac detected by immunofluorescence analysis in ESCs described in **f**. More than 200 cells are examined in each sample. **i** DNA integrity assessment of ESCs described in **f** by comet assay. More than 100 cells are examined in each sample. **j** Quantification of chromosomal breakage (left) or chromosome ends fusion (right) in ESCs as described in **f**. More than 200 cells are examined in each sample. *represents a difference from WT ESCs, #indicates a difference from TRIM66-KO + Ctrl. Data are presented as the means ± SEM. Statistical significance is determined by two-tailed Student's $t$-test. *#$p < 0.05$, **##$p < 0.01$, ***###$p < 0.001$, N.S., no significance. Source data are provided as a Source Data file

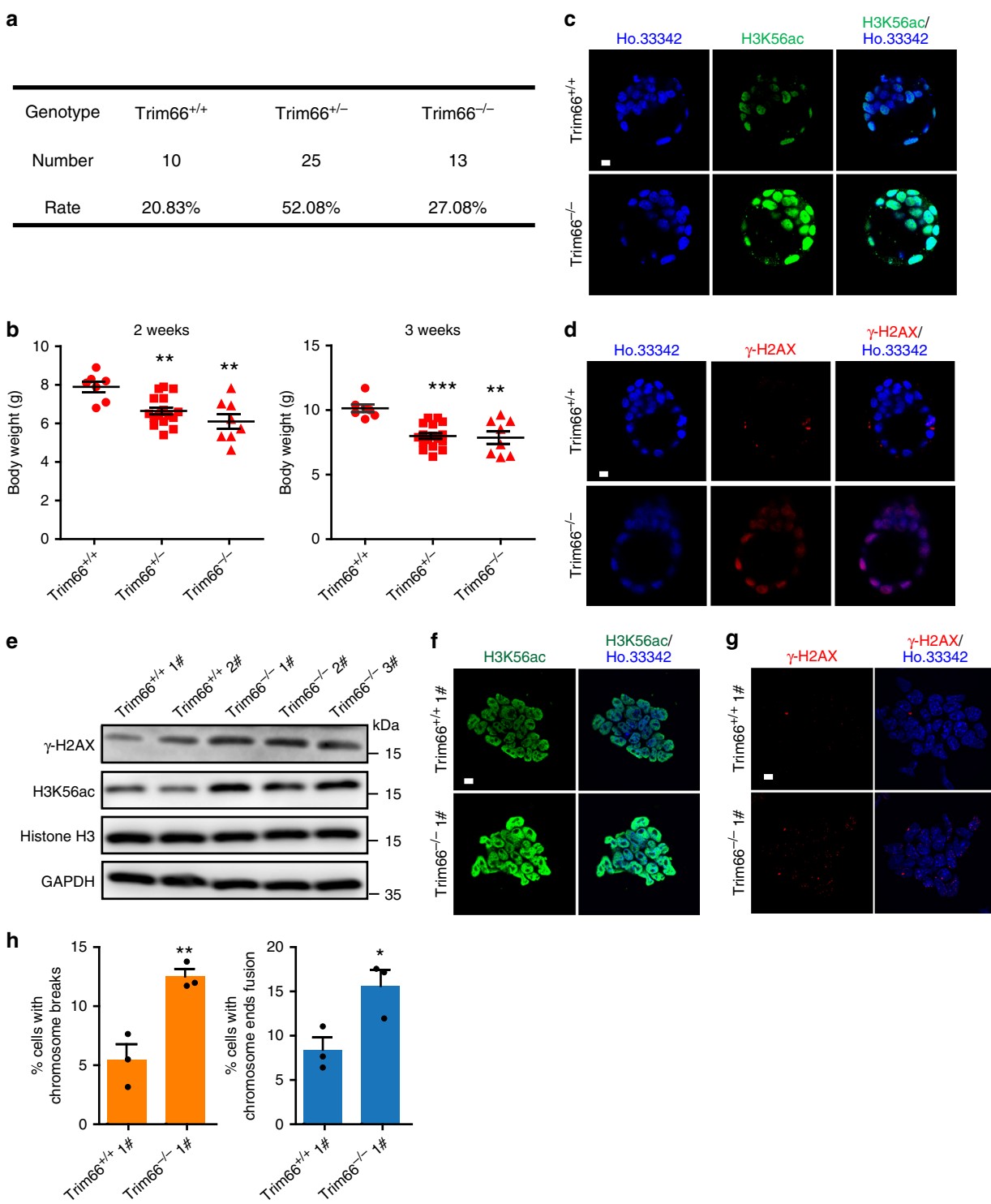

**Fig. 7** Loss of TRIM66 results in genomic instability in vivo. **a** Genotypic statistics of offspring mice. A total of 48 offspring after crossing 6 pairs of Trim66[+/−] mice. **b** Statistics of the body weight of mice offspring aged 2 or 3 weeks. **c** Representative immunofluorescence images of H3K56ac in blastocysts of WT Trim66[+/+] and Trim66[−/−] mice. Scale bar, 10 μm. **d** Representative immunofluorescence images of γ-H2AX in WT and Trim66[−/−] blastocysts. Scale bar, 10 μm. **e** Immunoblot analysis of the levels of γ-H2AX and H3K56ac in the primary Trim66[+/+] and Trim66[−/−] ESCs. **f, g** Representative immunofluorescence images of H3K56ac and γ-H2AX in the primary Trim66[+/+] and Trim66[−/−] ESCs. Scale bar, 10 μm. **h** Quantification of chromosomal breakage (left) or chromosome ends fusion (right) in primary Trim66[+/+] and Trim66[−/−] ESCs. More than 100 cells are examined in each sample. *represents a difference from Trim66[+/+] mice or primary Trim66[+/+] ESCs. Data are presented as the means ± SEM. Statistical significance is determined by two-tailed Student's t-test. *$p < 0.05$, **$p < 0.01$, ***$p < 0.001$. Source data are provided as a Source Data file

The nucleosome—the basic unit of chromatin—is composed of histone octamers and DNA. The histone octamer is wrapped by two superhelical turns of DNA, with an approximate length of 145–147 bp[68]. Considering the presence of H3K56 near the DNA entry/exit site of the nucleosome, we supposed that DNA enwinding the histones may prevent TRIM66 PHD-Bromodomain closing to H3K56ac, which may weaken the binding of TRIM66 PHD-Bromodomain with complete nucleosome. We supposed that there may be other protein factors that promote the binding of TRIM66 with nucleosome through interacting with TRIM66 and chromatin, which still needs further research. Moreover, our results showed that TRIM66 mediates the deacetylation of H3K56ac at the DSB sites. The loose winding of nucleosomal DNA nearby the DSB sites may expose the modification of H3K56ac and facilitate its recognition by TRIM66.

Together, we propose a model in which TRIM66 reads the specific H3R2me0K4me0-H3K56ac modifications and recruits Sirt6 to chromatin, to deacetylate the H3K56ac at the DSB sites. This process facilitates the subsequent recruitment of DNA repair proteins in ESCs. This study highlights the specific mechanism of the chromatin reader TRIM66 and H3K56ac in regulating DDR and genomic integrity. These data provide key clues and evidences for future studies aiming to elucidate the unique mechanisms of DDR in ESCs and deepen our understanding of histone PTMs in DDR and maintenance of genomic stability.

## Methods

**Plasmid constructions and bacteria strains**. The DNA fragments of human TRIM66$_{968-1160}$, TRIM24$_{824-1006}$, and TRIM33$_{882-1087}$ were amplified from a human bone marrow cDNA library using a PCR-based cloning strategy. The fragments of TRIM66-WT$_{968-1160}$ were inserted into a pGEX-4T-1 plasmid (GE Healthcare). The fragments of TRIM24$_{824-1006}$ and TRIM33$_{882-1087}$ were ligated into a pET-SUMO-28a (+) plasmid (Novagen). The DNA fragments of TRIM66-MUT$_{968-1160}$ and TRIM66-WT$_{965-1160}$ with codon optimization were synthesized by General Biosystems (sequence was shown in Supplementary Methods) and inserted into a pGEXT-4T-1 plasmid (GE Healthcare). Mutants of TRIM66 PHD-Bromo were generated through PCR using the Mutant Best kit (Takara). The mutation of point mutant based on the codon-optimized DNA sequence of TRIM66-WT$_{965-1160}$. All constructs were expressed in *Escherichia coli* BL21 (DE3). The construct of ASF1a$_{1-156}$ inserted into a pRSFDuet plasmid (courtesy of Ruiming Xu Lab, Institute of Biophysics, Chinese Academy of Sciences) was expressed in *E. coli* BL21 (DE3) RIL. The construct of full-length H4 inserted into a pET3a plasmid was expressed in *E. coli* BL21 (DE3). The expression of full-length histone H3K56ac was performed in BL21 (DE3) cells transformed with pBK-AcKRS3 and pET22b encoding H3 with amber codons at the K56 site[12]. The DNA fragment of full-length TRIM66 was synthesized by General Biosystems and ligated into fugw vector (Addgene). See the information of protein sequence in Supplementary Table 1.

Small guide RNAs targeting the exon 3 of all three Trim66 transcription variants were individually inserted into a pX330 vector. For the donor construction, 5′ arm segment, PGK promoter-Puro-3× polyA segment, and 3′ arm segment were cloned into a pLB vector (TIANGEN), and these plasmids were electroporated into ESCs using the Gene Pulser × Cell System (Bio-Rad). For the knockdown of Trim66, short hairpin RNAs targeting Trim66 were cloned into a pLKO.1 vector (Addgene) and a pLKO-Tet-On vector (courtesy of Xiaoqing Zhang Lab, Tongji University, China), respectively. For the overexpression of the WT, point mutations, and truncated mutations of TRIM66 and Sirt6 in ESCs and 293T cells, their cDNAs were cloned into a fugw vector (Addgene). All constructed plasmids were verified through DNA sequencing.

All primers are listed in Supplementary Tables 3 and 4.

**Expression and purification of proteins**. Protein expression (TRIM66$_{965-1160/968-1160}$, TRIM24$_{824-1006}$, and TRIM33$_{882-1087}$) was induced at A600 = 0.8 using 0.2 mM isopropyl β-d-1-thiogalactopyranoside (IPTG) and 100 μM ZnSO$_4$. ASF1a$_{1-156}$ protein was induced at A600 = 0.8 using 0.5 mM IPTG. Subsequently, the cells were incubated overnight at 16 °C for 24 h. The protein used for crystallographic and ITC studies was expressed in lysogeny broth medium. The uniformly [$^{15}$N]- and [$^{15}$N, $^{13}$C]-labeled samples used for NMR were expressed in LR medium (24 g L$^{-1}$ KH$_2$PO$_4$, 5 g L$^{-1}$ NaOH, 0.5 g L$^{-1}$ NH$_4$Cl, pH 7.0, 2.5 g L$^{-1}$ glucose, 100 μM CaCl$_2$, and 2.2 mM MgSO$_4$) with added glucose or [$^{13}$C$_6$] glucose and $^{15}$NH$_4$Cl.

TRIM66-WT$_{965-1160}$/TRIM66-MUT$_{968-1160}$ and its mutants were purified through glutathione 4B column chromatography (GE Healthcare) followed by thrombin cleavage (or eluted directly for GST-pulldown assays). These were further

purified through chromatography using a HiLoad Superdex 75 16/600 column (GE Healthcare) in a buffer containing 20 mM Tris, 1 M NaCl, and 5 mM dithiothreitol (DTT), pH 7.5. TRIM24$_{824-1006}$, TRIM33$_{882-1087}$, and ASF1a$_{1-156}$ were purified using a Ni-Chelating Sepharase Fast Flow column (GE Healthcare) followed by SUMO Protease. Digested protein was loaded onto a nickel column again to remove the His-sumo tag and the His-tagged protease. Further purification was achieved through chromatography using a HiLoad Superdex 75 16/600 column (GE Healthcare) (TRIM24$_{824-1006}$ and TRIM33$_{882-1087}$) or HiLoad Superdex 200 16/600 column (GE Healthcare) (ASF1a$_{1-156}$) in a buffer containing 20 mM Tris, and 0.5–1 M NaCl, pH 7.5. Histone H4 was induced at A600 = 0.8 using 0.5 mM IPTG and the cells were incubated at 37 °C for 6 h. Histone H3K56ac was supplemented with 20 mM nicotinamide, 10 mM acetyl-lysine (Kac), and 0.2% arabinose, and incubated for another 30 min before induction with 0.5 mM IPTG[12]. The purification procedure of histone was shown in Supplementary Methods.

**Peptide synthesis**. Peptide H3$_{1-12}$, H3$_{1-12}$K4me1, H3$_{1-12}$K4me2, H3$_{1-12}$K4me3, H3$_{9-19}$K14ac, H3$_{14-23}$K18ac, H3$_{18-28}$K23ac, H3$_{23-32}$K27ac, H3$_{31-40}$K36ac, H3$_{48-57}$K56ac, H4$_{1-10}$K5ac, H4$_{1-12}$K8ac, H4$_{7-17}$K12ac, H4$_{11-22}$K16ac, H4$_{15-25}$K20ac, H3$_{1-12}$R2me1, H3$_{1-12}$R2me2a, H3$_{1-12}$R2me2s, and H3$_{(1-15)-(48-57)}$ K56ac were used for the crystallization and binding assays. All peptides were synthesized by GL Biochem (Shanghai, China) and supplied with stringent analytical specifications (purity level >98%), which included high-performance liquid chromatography and mass spectrometry analysis. See the information of peptide sequence in Supplementary Table 1.

**Crystallization**. The mutant protein of the TRIM66 PHD-Bromodomain used for the crystallographic study contained residues 968 to 1160 with eight mutated residues (L1002T, C1026S, C1030S, Y1031H, M1036K, I1089T, C1135S, and V1138N). The mutant protein was dialyzed using a buffer containing 20 mM Tris, 100 mM NaCl, and 5 mM DTT at pH 7.5 prior to the crystallographic study. The H3$_{1-12}$ peptide was dissolved in the same buffer and adjusted to pH 7.5 with a final concentration of 20 mM. Crystals of the free-state TRIM66-PHD-Bromo mutant were obtained by mixing 8 mg ml$^{-1}$ protein with an equal volume of crystallization buffer (30% w/v 5000 MME, 100 mM Tris base/hydrochloric acid pH 8.0, and 200 mM lithium sulfate) using the hanging drop vapor diffusion method at 20 °C. Crystals of the complex of the mutant TRIM66-PHD-Bromo and unmodified H3$_{1-12}$ peptide were obtained by incubating 7.5 mg ml$^{-1}$ protein with peptide at a 1:3 ratio, followed by mixing with an equal volume of crystallization buffer (0.1 M HEPES sodium pH 7.5, 10% v/v 2-propanol, and 20% PEG4000) using the hanging drop vapor diffusion method at 20 °C.

**Data collection and structure determination**. The X-ray data sets for crystals of the mutant TRIM66-PHD-Bromo in the free state and complex of mutant TRIM66-PHD-Bromo and H3$_{1-12}$ were collected using beamlines 19U at the Shanghai Synchrotron Radiation Facility (100 K) and at wavelength of 0.97891 Å and 0.97776 Å, respectively. The data were processed using the program HKL2000 suite and programs in the CCP4 suite. The structure of the free-state TRIM66-PHD-Bromo mutant was determined by molecular replacement using PHASER and TRIM24-PHD-Bromo structure (PDB ID: 3O33)[6] as the search model[69]. The complex structure of TRIM66-PHD-Bromo and H3$_{1-12}$ was also determined by molecular replacement using PHASER and the free-state TRIM66-PHD-Bromo structure as the search model[69]. The model was further refined using COOT[70] and PHENIX[71]. The Ramachandran statistics showed that 98.1% (TRIM66-MUT$_{968-1160}$) and 99.5% (TRIM66-MUT$_{968-1160}$-H3$_{1-12}$) of all the residues were in the favored region and all the other residues were in the allowed region. The crystallographic statistics were listed in Table 1. All structure figures were prepared using PyMOL[72].

**ITC measurements**. The calorimetric experiments were performed at 25 °C using a MicroCal PEAQ-ITC. The proteins were dialyzed using a buffer containing 20 mM Tris and 100 mM NaCl at pH 7.5. Lyophilized peptides were dissolved in the same buffer and adjusted to pH 7.5 prior to use. Peptide concentrations were estimated from the quantitative NMR experiments[73,74]. ITC measurements were performed via injection of 0.7–4 mM peptide into a cell containing 0.12 mM TRIM24-PHD-Bromo, 0.05 mM TRIM33 PHD-Bromo, and 0.05 or 0.11–0.15 mM TRIM66-PHD-Bromo (or its mutants). The ITC data were subsequently analyzed using the MicroCal PEAQ-ITC analysis software.

**NMR spectroscopy and backbone assignment**. All NMR experiments were performed using a Bruker Avance III 850 MHz spectrometer equipped with a cryoprobe at 302.7 K.

For the sequential backbone assignment, a uniformly [$^{15}$N, $^{13}$C]-labeled TRIM66 protein was concentrated to 0.5 mM in buffer containing 10 mM Bis-Tris, 100 mM NaCl, and 5 mM DTT at pH 6.8. We adopted a non-uniform sampling method to enhance the sensitivity of all the NMR spectra[75,76]. Moreover, we applied a semi-automatic covariance LCC assignment protocol for the backbone assignment[77]. A total of 222 complex points were sampled for the indirect dimensions on a uniform 64 × 64 time grids using a randomized concentric shell

sampling pattern. High-sensitivity NMR spectra HN (CO) CA and HNCA were used to calculate the covariance spectrum and perform the backbone assignment, whereas few signals in the low-sensitivity spectra CBCA (CO) NH, CBCANH, HNCO, and HN (CA) CO were used to cross-validate the result of the assignment. All these spectra were preprocessed in NMRPipe using conventional parameters and subsequently reconstructed using the SCRUB software. Finally, all spectra were converted to the format of Sparky software or covariance LCC for further analysis.

For the NMR titration experiment, a uniformly $^{15}N$-labeled TRIM66 protein was concentrated to 0.5 mM in buffer containing 10 mM Bis-Tris, 100 mM NaCl, and 5 mM DTT at pH 6.8. The titration spectra were preprocessed in NMRPipe using routine parameters and subsequently converted to the format of Sparky for further analysis.

**GST-pulldown assays.** The recombination methods of the H3-H4 tetramer and the H3-H4-ASF1a complex in vitro were shown in Supplementary Methods[12,53]. About 50 µg protein, GST, GST-TRIM66-WT$_{965-1160}$, or its mutant (D986A, N1123A, and D986A/N1123A) were bound to glutathione-sepharose beads (GE Healthcare) in a buffer (20 mM Tris, 200 mM NaCl, and 1 mM DTT) at pH 7.5 and incubated at 4 °C for 4 h. The glutathione-sepharose beads were pelleted and washed thrice with washing buffer. Subsequently, the beads were incubated with 100–150 µg recombinant H3K56ac-H4-ASF1a complex in a buffer containing 20 mM Tris, 150 mM NaCl, and 1 mM DTT at pH 7.5 for 6 h at 4 °C. The beads were pelleted and washed thrice using the same buffer. The captured proteins were eluted and analyzed using SDS-polyacrylamide gel electrophoresis (PAGE).

For the GST-pulldown assays of the TRIM66-WT$_{965-1160}$ against the calf thymus histones, the beads with about 50 µg GST-TRIM66-WT$_{965-1160}$ were incubated with 50 µg calf thymus histones in a buffer containing 20 mM Tris, 500 mM NaCl, and 1 mM DTT at pH 7.5 for 1 h at 4 °C. The beads were pelleted and washed five times using the same buffer. The captured proteins were eluted and analyzed using SDS-PAGE.

**Cell culture.** Mouse ESCs (E14.1) were maintained in Dulbecco's modified Eagle's medium (DMEM) supplemented with 15% (v/v) fetal bovine serum (Gibco), 0.1 mM nonessential amino acids (Gibco), 2 mM Glutamax (Gibco), 1 mM sodium pyruvate (Gibco), and 55 µM β-mercaptoethanol (Gibco), with 1000 units/ml leukemia inhibitory factor (Millipore) using 0.1% gelatin-coated (Millipore) plates. For the culture of ESCs, the medium was changed every day and the cells were passaged every 2 days. The 293T cells were cultured in DMEM containing 10% fetal bovine serum.

**Reverse-transcription and quantitative PCR.** Total RNA was extracted from the cells using RNAiso (Takara, 9109). First-strand cDNA synthesis was performed using the PrimeScript$^{TM}$ RT reagent kit (Takara, RR037A). The obtained cDNAs were analyzed through quantitative PCR (qPCR) with SYBR® Premix Ex Taq$^{TM}$ (Takara, RR420A) using an Agilent Stratagene Mx3000P instrument. The primer sequences used in this study for qPCR are listed in Supplementary Table 5.

**Immunoblotting analysis.** The cells were washed in 1× phosphate-buffered saline (PBS) and lysed in SDS lysis buffer containing 1 × PhosSTOP protease inhibitor (Roche) and a protease inhibitor cocktail (Roche). Equal amounts of the cell lysates were separated using SDS-PAGE and blotted onto polyvinylidene fluoride membranes (Millipore). Nonspecific binding was blocked through incubation with 3% bovine serum albumin at room temperature for 1 h. Subsequently, blots were probed with primary antibodies at 4 °C, followed by incubation with appropriate secondary antibodies. Signals were visualized using enhanced chemiluminescence (Bio-Rad). The antibodies used in this study are listed in Supplementary Table 6.

**Co-immunoprecipitation.** ESCs or 293T cells (5 million) in 60 mm dishes were co-transfected with 2.5 µg Flag-Sirt6 and 2.5 µg WT, point mutated, or truncated HA-TRIM66 for 48 h. Subsequently, the cells were washed in 1× PBS and lysed in 400 µl IP buffer consisting of Lysis Buffer I and Lysis Buffer II at a ratio of 1:2 (Lysis Buffer I: 450 mM NaCl, 40 mM Tris pH 7.4, 1% Triton X-100, 20% glycerol, and 1 mM EDTA; Lysis Buffer II: 10 mM NaCl, 40 mM Tris pH 7.4, 1% Triton X-100, and 20% glycerol). Cells were sonicated for 30 s (with a pause of 2 s for every 1 s of sonication) at 25% Amp (QSNOICA Sonicators). The cell lysates were incubated with antibodies and control IgG at 4 °C, respectively. The mixture of Ezview Red Protein A Affinity Gel (Sigma-Aldrich) and Ezview Red Protein G Affinity Gel (Sigma-Aldrich) (1:1) was incubated with the antibody-containing lysates for 4 h at 4 °C. Subsequently, the protein-bead complex was washed with 400 µl IP buffer thrice, followed by heating at 95 °C for 10 min in 1× SDS buffer for western blotting analysis.

**Comet assay.** Comet assays were performed according to the protocol of the Trevigen Comet Assay® Kit. Briefly, cells were washed with 1 × PBS (without Ca$^{2+}$ and Mg$^{2+}$) thrice and resuspended. The cell suspension was mixed with Comet LMAgarose and spread on the surface of the Comet Slides (50 µl per slide). The LMAgarose on the slides were solidified at 4 °C for 30 min, immersed in lysis solution at 4 °C for 30 min, and soaked in alkaline unwinding solution at room temperature for 30 min. Subsequently, the slides with LMAgarose were electrophoresed using the Comet Assay® ES system and stained with SYBR Green I.

**Immunofluorescent staining.** For the immunofluorescent staining of ESCs and blastocysts, the cells were fixed with 4% paraformaldehyde (PFA) at room temperature for 15 min followed by permeabilization with 0.2% Triton X-100 in PBS for 8 min. The cells were blocked with 10% donkey serum in PBS for 1 h at room temperature. Subsequently, the cells were incubated with primary antibody (at 4 °C overnight) followed by AlexaFluor-conjugated secondary antibody incubation (Invitrogen) for 1 h and counterstained using Hoechst33342. Fluorescent images were captured using a Nikon A1R confocal microscope (Nikon).

**Chromatin isolation.** Cells were washed in PBS and lysed in Buffer A (1.5 mM MgCl$_2$, 10 mM KCl, 10 mM HEPES, 0.34 M sucrose, 10% glycerol, and 0.1% Triton X-100) containing 1 mM DTT, 1 mM Na$_3$VO$_4$, and a protease inhibitor cocktail (Roche) on ice for 30 min. After centrifugation at $1700 \times g$ for 5 min, the remaining pellets were resuspended in Buffer B (3 mM EDTA, 0.3 mM EGTA) containing 1 mM DTT, 1 mM Na$_3$VO$_4$, and a protease inhibitor cocktail, followed by incubation on ice for 30 min. After centrifugation at $5000 \times g$ for 5 min, the pellets were solubilized in SDS sample buffer and analyzed as the chromatin fraction through western blotting.

**Chromosome metaphase spread preparation.** Cells were incubated with 0.2 µg/ml colcemid (Sigma) for 4 h to enrich the cells at the metaphase. The enriched cells were collected through trypsinization, resuspended in 75 mM KCl at 37 °C for 5 min, and fixed with methanol/glacial acetic acid (3:1) at room temperature for 1 h. Subsequently, the cells were spread onto clean slides. Images were captured using a Nikon A1R microscope (Nikon).

**Alkaline phosphatase (AP) staining.** AP staining was performed using the FastRed Alkaline Phosphatase Kit (Sigma). ESCs were fixed in ice-cold 4% PFA for 1 min. ESCs were subsequently washed and incubated with staining solution for 15 min at room temperature followed by examination under the microscope to capture images.

**Animal studies and collection of mouse embryos.** Trim66$^{+/-}$ mice were provided by Professor Dahua Chen. All mice were maintained in a pathogen-free environment throughout the experiments. We have complied with all relevant ethical regulations for animal testing and research. All animal experiments were approved by the Institutional Animal Care Committee of Tongji University.

For the collection of embryos at the blastocyst stage, female mice (8–10 weeks old) were superovulated via injection with 10 IU of pregnant mare serum gonadotropin, followed by injection with 10 IU of human chorionic gonadotropin (San-Sheng Pharmaceutical, Co. Ltd) 48 h later. The superovulated female mice were mated with male mice. Subsequently, the embryos at the blastocyst stage were collected from the uterus of pregnant mice at E3.5.

All blots and gels are provided in the Source Data.

**Statistical analysis.** Statistical analyses were performed using the GraphPad Prism 5 (GraphPad Software, Inc., La Jolla, CA, USA). Data are presented as the means ± SEM from three independent experiments ($n = 3$); $^*/^\#p < 0.05$, $^{**}/^{\#\#}p < 0.01$, and $^{***}/^{\#\#\#}p < 0.001$, respectively (two-tailed Student's $t$-test). Statistical parameters for each experiment, including values of $n$ and statistical significance, are shown in the corresponding figure legends.

**Reporting summary.** Further information on research design is available in the Nature Research Reporting Summary linked to this article.

## Data availability

Atomic coordinates and structure factors have been deposited in the Protein Data Bank (https://www.rcsb.org/), with accession numbers 6IET (TRIM66-MUT$_{968-1160}$) and 6IEU (TRIM66-MUT$_{968-1160}$-H3$_{1-12}$). All additional data that support the findings of this study are available from the corresponding author upon reasonable request. Uncropped gel images and other source data underlying Figs. 3a–c, e, g, 4c, e–h, 5a–c, e, g–j, 6a–c, e–j, 7b, e, h, Supplementary Figs. 1e, 2a, 8b, d, g, 9a, d, e, 10a–e, 12a–c, e–g, 13 are provided as a Source Data file. A reporting summary for this article is available as a Supplementary Information file.

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

## Acknowledgements

We thank Dahua Chen for providing us *Trim66*[+/−] mice. We thank Jiahai Zhang, Hongbin Sun, and Junfeng Wang for the assistance with NMR work. Part of our NMR work was performed at High Magnetic Field Laboratory, Chinese Academy of Sciences. This work was supported by Ministry of Science and Technology of China [2016YFA0500700 and 2016YFA0101300]; the Strategic Priority Research Program of the Chinese Academy of Science [XDPB10, XDB08010101]; Chinese National Natural Science Foundation [31330018, 31500590, 81530042, 31830059, and 31701110]; Fundamental Research Funds for the Central Universities [22120190149].

## Author contributions

J.J.C., Z.K.W., J.H.K. and Y.Y.S. conceived the experiments. J.J.C. performed binding assays, crystallography, and NMR experiment under the guide of F.D.L. and D.S.G. Z.K.W. performed the phenotypic and mechanistic analyses under the guide of X.D.G. Q.T.W. is responsible for the sequential backbone assignment. X.W.C. performed part of the ITC assays. Y.X.X. and W.C. guided part of plasmid construction. J.J.C. and Y.R.L. purified the histones. J.J.C. and Z.K. W generated the figures. J.J.C., Z.K.W., X.D.G., J.H.K. and Y.Y.S. wrote the manuscript.

## Additional information

**Competing interests:** The authors declare no competing interests.

