## [Peer Review File · Nature Communications]

Reviewers' comments:

Reviewer #1 (Remarks to the Author):

The manuscript by Chen et al. identifies the bromodomain protein TRIM66 as a reader of H3R2K4 and H3K56Ac, as well as a DNA damage factor. This work obtains a crystal structure of the PhD-BRD domain of TRIM66 and using ITC experiments, show that this protein binds to unmodified H3R2K4 and H3K56Ac. Cell-based experiments revealed that deletion or deficiency of TRIM66 resulted in aberrant DNA damage signaling and induced signs of genome instability. A mouse KO line was generated and mutant mice were smaller and cells exhibited increased DNA damage markers and H3K56Ac. Biochemical and molecular biology experiments place TRIM66 as a regulator of the HDAC SIRT6, which has been linked to regulation of H3K56Ac and DNA damage signaling previously. Overall, this study provides evidence for a role of TRIM66 in protecting genome integrity. While this study is of potential interest given the importance of chromatin regulators in mediating DNA damage signaling and repair, as well as the potential involvement of these proteins in human diseases, there are several issues that reduce the enthusiasm of this reviewer for this work.

Main Issues:

1. For the structural studies, no data is provided for the proteins that are being used and typical validation experiments using peptides and purified proteins are not provided. It is unclear even how these marks were identified as targets of TRIM66. As many mutations were needed to obtain a protein that behaved for these studies, these results should be validated from wt proteins from cells using peptide pull-down experiments. Is it even possible that an individual polypeptide would be able to bind the N-terminal tail of H3 and H3K56Ac, which is located some distance from R2 and K4 of H3. Would this perhaps be done by a dimer or multimeric species of TRIM66? Modeling or additional experiments should address this issue.

2. For DNA damage signaling and repair, DNA repair and the biological significance of defects in signaling that occur in TRIM66 mutant cells are not provided. For example, what type of DNA repair does TRIM66 promote, do cells require TRIM66 to survive DNA damage, is TRIM66 directly bound at DNA damage sites and if so, how is this regulated. As it stands, these results may be indirect for TRIM66. The IF results for K56Ac in Fig. 5d are not sufficient to be able to claim anything about these reactions at DNA damage sites. Additional evidence needs to be provided. This is very little functional data for how important TRIM66 is for DNA repair and signaling so more evidence should be provided to support this main claim of the manuscript.

3. As KO mice were generated, the main thesis of this work that TRIM66 promotes DNA damage response and genome stability should be shown using this reagent, at least cells derived from these mice. In addition, the nomenclature provided for the karyotypes is incorrect. Using metaphase spreads, one cannot observe anaphase bridges as these occur post metaphase, the time when these cells are arrested for the analysis (ex. Fig. 3). Better images and description of these experiments is warranted.

Minor issues

1. For histone modification blots, the unmodified histone needs to be provided as a loading control.

2. There are several spelling mistakes throughout so some editing should be performed on the manuscript.

Reviewer #2 (Remarks to the Author):

This is one of those rare manuscripts in which the authors conduct a broad, yet rigorous analysis of molecular interactions and then establish their role in cell-based functional studies and even in whole animals. In this remarkable work, the authors set out to understand the molecular target of TRIM66, establishing through rigorous quantitative biophysics that the protein recognizes the unmodified amino terminus of histone H3, and contrasts this with behavior of other TRIM family members. The authors utilize the resulting information to dissect the regulatory network and interactome for TRIM66 in an impressive array of studies.

This is an outstanding paper, but it is not without its weaknesses. For one, this reviewer thinks that the authors over-reach and cram too much in this paper. For example, the data on interactions with SIRT6 could be stronger and the result itself is sufficiently important that it should be strengthened and packaged into a different paper. In addition, the authors dive very far into the weeds of individual hydrogen bonding networks, which is interesting to some readers, but will make it difficult for most readers to stay focused on the main ideas. Finally, there are weaknesses in some of the experimental data shown (described below). But in all, this is a fine paper, including the fascinating later part on murine development, which should be retained should the authors shorten the manuscript. Specific comments (some positive and negative) are below.

1. From the introductory background commentary, it is not entirely clear to the reader why the authors focused on TRIM66 and why they were so sure, from the beginning, that it recognizes the acetylation on H3K56. It seems as if certain prior work may not have been cited.
2. Page 6 - To the author's credit, in trying to figure out the TRIM66 code, they learned a lot about the broader TRIM protein family members and the basis for their target recognition. This adds a lot of value to the paper. If it were my paper, I would expand on this and then remove other parts (such as the SIRT6).
3. There is too much key data in the supplemental figures. This is another symptom of the fact that the paper is too big - needs to be broken up.
4. The NMR data add a lot to this already rigorous manuscript. The technical aspects of combining crystallography, NMR and ITC are outstanding and make for an unusually definitive body of work. This raises the bar for the field.
5. On page 8, regarding aggregation of the peptide-induced protein aggregation: didn't the aggregation problem undermine the experiment? How is the titration interpretable if this is going on? The reader was confused about why this information was in the paper, and whether there were subsequent problems with the data
6. On page 9, given the authors' claims, this reviewer would have liked to see a more direct measure of the upregulated expression of H2AX. The immunoblot is helpful, but it is very qualitative. Other types of direct metrics, along with QRT-PCR would have been important to include.
7. In the text, the authors need to be more clear about what is KO-1 and what is KO-2 in Figure 3. The text does not refer effectively to figure 3. In general in this paper, the main text does a poor job referring to the figures, which are not labeled consistently with the language in the text.
8. This reviewer is confused by the immunoblot data or perhaps the labeling in Fig. 3 and 5 (as in 5a). The left and right panels differ in the amount of dox exposure (48 vs. 96h), but the control lanes look different under the same conditions (where dox is on in both cases). The layout of these figures was confusing, and the effects did not seem pronounced.

9. The evidence for an interaction with Sirt6 was not particularly strong. The co-IP could simply illustrate indirect association within a larger complex. Particularly given the biophysical experiments at the beginning of this paper, if the author hopes to stretch the study this far, they should prove a direct interaction between TRIM66 and Sirt6 to make their assertions.

10. There is too much in this paper - the Sirt6 should be done more rigorously, with some added biophysical binding data, and put into a different paper.

11. However, this reviewer really liked the data on TRIM66 and murine development (Fig. 7). This was a very nice addition to the paper.

Reviewer #3 (Remarks to the Author):

Chen et al determined the crystal structure of the PHD-Bromo tandem domains of TRIM66, free and in complex with unmodified H3 tail. This structural study, together with ITC assays, reveals a specific recognition of the unmodified H3 tail by the PHD-Bromo domains. Through NMR titration, they also identified an interaction between the Bromodomain and the H3(48-57)K56Ac peptide. Their cellular analysis indicated that the TRIM66-histone recognition mediates the K56Ac deacetylation and is important for DNA damage repair and genomic stability in ES cells. Their cellular studies further revealed an interaction between TRIM66 and deacetylase Sirt6, and that such an interaction mediates the deacetylation of H3K56Ac and initiation of DDR. Overall, this study is of interest to understanding the functional role of TRIM66 and the regulatory mechanisms of H3K56Ac and DDR. However, I find that the structure-function connection needs to be strengthened.

1. The authors did not provide any biochemical/cellular evidence to support the claim that "We didn't find any binding partners of the PHD-Bromodomain from the acetylated H3 and H4 peptides except the acetylated H3K56 peptide." Such evidence, such as systematic histone array or pull-down assays, would be important for confirming the binding specificity of the Bromodomain.

2. It is unclear why the authors chose to use the H3(48-57)K56Ac peptide for the binding assay. Given that H3K55 is located at the C-terminal end of the helix 1, one would argue that the binding assay with the H3(48-57)K56Ac peptide, which mostly lacks secondary structure, would not be sufficient to capture the interaction, if any.

3. Relatedly, the ITC data for the K56Ac-TRIM66 binding gave a stoichiometric ratio that significantly deviates from 1 (Figure 1d and Figure S3b), which appears to suggest that the observed heat change may not dominantly arise from the specific binding. Could the K56Ac-induced TRIM66 aggregation, as the authors proposed, contribute to the heat change? How were those peptides containing no Tyr nor Trp quantified for the ITC measurement? Were the errors of the ITC parameters estimated from multiple independent experiments or the curve fitting of one single experiment?

4. Evidence on a direct interaction between TRIM66 and H3K56ac in cells is also missing. An immunofluorescence assay on the cellular co-localization of H3 K56Ac with TRIM66 wild-type and mutants would help strengthen the link.

5. English in the text needs to be improved. There are numerous grammatical issues. To list a few:

In the first paragraph, page 2: The sentence "The exact biological function of TRIM66 and its ability to read specific PTMs and execute its biological roles remains unknown" is confusing and needs to be clarified.

In the second paragraph, page 2: Change "are existed" to "exist".

In the third paragraph, page 2: Change "non-histone proteins, which is..." to "non-histone proteins, which are...".

In the first paragraph, page 3 and many other places, change "N-terminal of histone H3" to "N-terminus of histone H3".

In the first paragraph, page 5, remove "with" from "contacted with"

In the second paragraph, page 5, change "This finding was consistent" to "This finding is consistent".

Supplementary Table 5. Change "not detectable binding" to "no detectable binding".

In Methods, change "The DNA fragments of human TRIM66, TRIM24 and TRIM33" to "The DNA fragment of human TRIM66, TRIM24 and TRIM33".

In Methods, change "ASF (1-156) into a pRSFDuet" to "ASF (1-156) inserted into a pRSFDuet".

Point-by-point response

Reviewer #1 (Remarks to the Author):

The manuscript by Chen et al. identifies the bromodomain protein TRIM66 as a reader of H3R2K4 and H3K56Ac, as well as a DNA damage factor. This work obtains a crystal structure of the PHD-BRD domain of TRIM66 and using ITC experiments, show that this protein binds to unmodified H3R2K4 and H3K56Ac. Cell-based experiments revealed that deletion or deficiency of TRIM66 resulted in aberrant DNA damage signaling and induced signs of genome instability. A mouse KO line was generated and mutant mice were smaller and cells exhibited increased DNA damage markers and H3K56Ac. Biochemical and molecular biology experiments place TRIM66 as a regulator of the HDAC SIRT6, which has been linked to regulation of H3K56Ac and DNA damage signaling previously. Overall, this study provides evidence for a role of TRIM66 in protecting genome integrity. While this study is of potential interest given the importance of chromatin regulators in mediating DNA damage signaling and repair, as well as the potential involvement of these proteins in human diseases, there are several issues that reduce the enthusiasm of this reviewer for this work.

Main Issues:

1. For the structural studies, no data is provided for the proteins that are being used and typical validation experiments using peptides and purified proteins are not provided. It is unclear even how these marks were identified as targets of TRIM66. As many mutations were needed to obtain a protein that behaved for these studies, these results should be validated from wt proteins from cells using peptide pull-down experiments.

Response:

Thanks for the reviewer's kind suggestions. We have provided the data of the proteins in the revised Table S1 and Fig. S1e.

As we have added in line 30-33 in revised manuscript, "**Both PHD finger domain and Bromodomain are evolutionarily and structurally conserved module. The majority of canonical single PHD finger domains were reported to read the N-terminal tail of histone H3, mainly the methylation status of H3K4(H3K4me2/3 or H3K4me0). While the Bromodomains were found to read the acetylated lysine, especially that on histone H3 and H4.**" We first perform the GST pulldown assays with calf thymus histones, which showed that TRIM66 PHD-Bromodomain can interact with histones, mainly H3 and H4 (Fig. S2a). According to previous studies, sequence alignment, structure comparisons and the results of GST pulldown, we performed a series of interactional studies between TRIM66 PHD-Bromodomain and histone peptides (unmodified N terminus of H3, H3K14ac, H3K18ac, H3K23ac, H3K27ac, H3K36ac, H3K56ac, H4K5ac, H4K8ac, H4K12ac, H4K16ac, H4K20ac) derived from H3 and H4 by ITC (description was added in line 109-127 in revised manuscript, data was provided in Fig. 1d, S2, S3). We found that TRIM66 PHD-Bromodomain can bind the N terminus of H3 and H3K56ac.

In the original manuscript, we didn't describe clearly about the name of the proteins using in different experiments. Thanks for your kind suggestions, we have clarified them now in the revised manuscript (e.g., line 97, line 189) and listed it in Table S1 accordingly. In fact, we used the *wt*

protein (TRIM66-WT₉₆₅₋₁₁₆₀) in all the ITC and GST pull down interaction assays. Meanwhile, the TRIM66-GSGS₉₆₈₋₁₁₆₀ and the TRIM66-MUT₉₆₈₋₁₁₆₀ was only used in the NMR titration experiments and crystal experiment, respectively. We observed that the binding affinity of TRIM66-MUT₉₆₈₋₁₁₆₀ ($K_D = 19.00 \mu\text{M}$) with H3₁₋₁₂ (Table S3) was nearly equal to the *wt* protein ($K_D = 15.70 \mu\text{M}$) and the binding affinity of TRIM66-GSGS₉₆₈₋₁₁₆₀ ($K_D = 127.00 \mu\text{M}$) with H3K56ac peptide (Table S3) was also nearly equal to the *wt* protein ($K_D = 109.00 \mu\text{M}$).

2. Is it even possible that an individual polypeptide would be able to bind the N-terminal tail of H3 and H3K56Ac, which is located some distance from R2 and K4 of H3. Would this perhaps be done by a dimer or multimeric species of TRIM66? Modeling or additional experiments should address this issue.

Response:

Although H3K56ac is located some distance from H3R2K4, the residues between H3R2K4 and H3K56 are mostly unstructured and very flexible (Chen et al., 2015; Gaubert et al., 2015; Luger et al., 1997; Natsume et al., 2007). The flexible H3 N terminal tail makes it possible that one TRIM66 PHD-Bromodomain binds H3R2K4 and H3K56ac.

We first performed a static light scattering experiments of the TRIM66-WT₉₆₅₋₁₁₆₀ and TRIM66-GSGS₉₆₈₋₁₁₆₀ protein. The results (TRIM66-WT₉₆₅₋₁₁₆₀, 23.19 kDa (actual value: 22.85 kDa) and TRIM66-GSGS₉₆₈₋₁₁₆₀, 22.76 kDa (actual value: 21.65 kDa)) showed that these two proteins are monomer in solution (see figures below).

We have showed that the TRIM66 PHD-Bromodomain binds H3₍₁₋₁₅₎₋₍₄₈₋₅₇₎ K56ac peptide in a combinational manner (Fig. 1d). To further figure out the binding mode between the TRIM66 PHD-Bromodomain and the united H3₍₁₋₁₅₎₋₍₄₈₋₅₇₎ K56ac peptide, we performed a NMR titration experiment and carefully analyzed the linewidth changes of TRIM66-GSGS₉₆₈₋₁₁₆₀ (this protein is feasible for NMR linewidth analysis and binds H3₍₁₋₁₅₎₋₍₄₈₋₅₇₎ K56ac peptide with an affinity(0.57 μM) nearly equal to the TRIM66-WT₉₆₅₋₁₁₆₀ (0.52 μM)) NMR signals along the ¹H dimension, which is sensitive to protein molecular weight changes. We found that the linewidth of the residues closed to the binding interface broaden significantly (larger than the digital resolution of 7 Hz, experiments were executed on a NMR spectrometer Bruker 500MHz) at titration ratio 1:1, consistent with the binding affinity from ITC assays. Meanwhile, we also found many residues located far from the binding interfaces only showed tiny linewidth changes (smaller than 7 Hz). Remarkably, except for the residues located at the flexible regions, whose linewidth was insensitive to molecular weight

changes, we also found many residues at the rigid region (e.g., W1143, L1114, K1093, S995, L1136, L1055, H997, L1144, E1141, shown in table below), showing that the molecular weight didn't change significantly after titrated with the united peptide H3₍₁₋₁₅₎₋₍₄₈₋₅₇₎ K56ac at ratio 1:1.

These results indicate that the combinational peptide binds with a stochastic ratio of 1:1 to TRIM66-GSGS₉₆₈₋₁₁₆₀, and both of the N-terminus of H3 and H3K56ac region of the united peptide can bind to the targeted pocket of the TRIM66-GSGS₉₆₈₋₁₁₆₀.

Residue name	Linewidth (Hz) along ¹ H dimension without peptide	Linewidth (Hz) along ¹ H dimension with equal molar peptide
W1143	36.8	37.5
L1114	28.3	29.3
K1093	22.4	25.5
S995	27.1	30.3
L1136	51.6	51.9
L1055	28.5	28.8
H997	29.6	27.3
L1144	38.8	41.6
E1141	32.0	36.7

References:

- Chen S, Rufiange A, Huang H et al., Structure–function studies of histone H3/H4 tetramer maintenance during transcription by chaperone Spt2. *Genes & Development*. 2015 29, 1326-1340.
- Gaubert A, Besle A, Guichard B et al., Structural insight into how the human helicase subunit MCM2 may act as a histone chaperone together with ASF1 at the replication fork. *Nucleic acids research*. 2015 43, 1905-1917.
- Luger K, Mäder A, Richmond R et al., Crystal structure of the nucleosome core particle at 2.8 Å resolution. *Nature*. 1997 389, 251.
- Natsume R, Eitoku M, Akai Y et al., Structure and function of the histone chaperone CIA/ASF1 complexed with histones H3 and H4. *Nature*. 2007, 446, 338.

3. For DNA damage signaling and repair, DNA repair and the biological significance of defects in signaling that occur in TRIM66 mutant cells are not provided. For example, 1) what type of DNA repair does TRIM66 promote ? 2) do cells require TRIM66 to survive DNA damage ? 3) is TRIM66 directly bound at DNA damage sites and if so, how is this regulated. As it stands, these results may be indirect for TRIM66. The IF results for K56Ac in Fig. 5d are not sufficient to be able to claim anything about these reactions at DNA damage sites. Additional evidence needs to be provided. This is very little functional data for how important TRIM66 is for DNA repair and signaling so more evidence should be provided to support this main claim of the manuscript.

Response:

Thanks for your thoughtful suggestions. We performed cell counting and CCK8 assays and our results showed that TRIM66 depletion significantly impaired the cell growth of ESCs under etoposide (EPI) treatment, suggested that TRIM66 KO ESCs became sensitive to DNA damage

reagent (Fig. S9b, c). We further performed the competitive survival experiment through co-culturing the GFP-labeled wild-type ESCs and TRIM66 KO ESCs under EPI treatment, and our results showed that the ratio of TRIM66 KO ESCs was gradually decreased, suggested that TRIM66 was indeed required for ESC to survive DNA damage (Fig. S9d). Our in-depth study found that TRIM66 depletion resulted in serious DNA damage signal and significantly abolished the co-localization of γ -H2AX foci and HR repair related proteins (Rad51 and CtIP) or NHEJ repair related protein (DNA-PKcs) in ESCs (Fig. 5f, 5g, S11d, and S11e). We further applied two specific inhibitors (YU238259 and NU7441) to block HR and NHEJ repair respectively (Stachelek et al., 2015; Robert et al., 2015), and these findings showed that TRIM66-depleted ESCs presented a significant decrease in the survival rate of ESCs under EPI treatment, suggested that TRIM66 could affect both HR and NHEJ repair pathways in ESCs (Fig. S11f). To better clarify the mechanisms of TRIM66 involved in the regulation of DNA damage, we performed the immunofluorescence assay and our results showed that Myc-tagged TRIM66 could co-localize with γ -H2AX foci and was able to bind at DNA damage sites (Fig. S11a). H3K56ac is localized at the globular domain of core H3 and can be rapidly deacetylated at the DNA damage sites, which is a prerequisite for the subsequent DNA damage repair (Battu et al., 2011; Tjeertes et al., 2009). We also analyzed the relevance of H3K56ac and γ -H2AX foci via Image J (Tian et al., Cell. 2016) and our findings indicated that the depletion of TRIM66 disrupted the negative correlation of H3K56ac and γ -H2AX (Fig. 5d). In view of the recognition of TRIM66 and H3K56ac (Fig.2 and Fig. S10), we speculated that the loose winding of nucleosomal DNA nearby the DNA damage sites might expose H3K56ac to its reader TRIM66, while TRIM66 deficiency impeded the erasure of H3K56ac at DNA damage sites and the recruitment of DNA repair proteins.

References:

- Stachelek G, Peterson-Roth E, Liu Y et al., YU238259 is a novel inhibitor of homology-dependent DNA repair that exhibits synthetic lethality and radiosensitization in repair-deficient tumors. *Molecular Cancer Research*, 2015 13, 1389-1397.
- Robert F, Barbeau M, Ethier S et al., Pharmacological inhibition of DNA-PK stimulates Cas9-mediated genome editing, *Genome Med*, 2015 7, 93.
- Battu A, Ray A, and Wani A. ASF1A and ATM regulate H3K56-mediated cell-cycle checkpoint recovery in response to UV irradiation. *Nucleic Acids Res*. 2011 39, 7931-7945.
- Tjeertes J, Miller K, and Jackson S. Screen for DNA-damage-responsive histone modifications identifies H3K9Ac and H3K56Ac in human cells. *The EMBO Journal* 2009 28, 1878-1889.
- Tian Y, Garcia G, Bian Q et al., Mitochondrial stress induces chromatin reorganization to promote longevity and UPR^{mt}. *Cell*, 2016 165, 1197-1208.

4. As KO mice were generated, the main thesis of this work that TRIM66 promotes DNA damage response and genome stability should be shown using this reagent, at least cells derived from these mice. In addition, the nomenclature provided for the karyotypes is incorrect. Using metaphase spreads, one cannot observe anaphase bridges as these occur post metaphase, the time when these cells are arrested for the analysis (ex. Fig. 3). Better images and description of these experiments is warranted.

Response:

Thanks for your suggestions. We further derived the primary ESCs from Trim66 KO mice and found that the primary Trim66-depleted ESCs exhibited elevated γ -H2AX and H3K56ac levels (Fig. 7e-g), consistent with these previous results in the isolated blastocysts and mouse ESC line. Similarly, a higher rate of chromosomal breakage and chromosome ends fusion was also observed in the primary Trim66-depleted ESCs (Fig. 7h). We apologized for misusing the nomenclature of anaphase bridges and have corrected into “chromosome ends fusion” in the revised manuscript and figure 3g, 4h, 6j, 7h, and S8e. Besides, we have provided better images of karyotype as shown in Fig. 3f.

Minor issues

1. For histone modification blots, the unmodified histone needs to be provided as a loading control.

Response:

We have provided the unmodified histone as a loading control as shown in Fig. 5a, 5h, 6f, and 7e.

2. There are several spelling mistakes throughout so some editing should be performed on the manuscript.

Response:

We have corrected the spelling mistakes in the revised manuscript and edited the entire manuscript carefully.

Reviewer #2 (Remarks to the Author):

This is one of those rare manuscripts in which the authors conduct a broad, yet rigorous analysis of molecular interactions and then establish their role in cell-based functional studies and even in whole animals. In this remarkable work, the authors set out to understand the molecular target of TRIM66, establishing through rigorous quantitative biophysics that the protein recognizes the unmodified amino terminus of histone H3, and contrasts this with behavior of other TRIM family members. The authors utilize the resulting information to dissect the regulatory network and interactome for TRIM66 in an impressive array of studies. This is an outstanding paper, but it is not without its weaknesses. For one, this reviewer thinks that the authors over-reach and cram too much in this paper. For example, the data on interactions with SIRT6 could be stronger and the result itself is sufficiently important that it should be strengthened and packaged into a different paper. In addition, the authors dive very far into the weeds of individual hydrogen bonding networks, which is interesting to some readers, but will make it difficult for most readers to stay focused on the main ideas. Finally, there are weaknesses in some of the experimental data shown (described below). But in all, this is a fine paper, including the fascinating later part on murine development, which should be retained should the authors shorten the manuscript. Specific comments (some positive and negative) are below.

1. From the introductory background commentary, it is not entirely clear to the reader why the authors focused on TRIM66 and why they were so sure, from the beginning, that it recognizes the acetylation on H3K56. It seems as if certain prior work may not have been cited.

Response:

There are only four proteins containing PHD-Bromodomain in TRIM family. Though the structure and function of PHD-Bromodomain of TRIM24, TRIM28 and TRIM33 have been studied well, little is known about the TRIM66 PHD-Bromodomain. So we focused on the TRIM66 to try to figure out the structure and function of it. Thanks for the reviewer's suggestions, we have added the descriptions why we focus on TRIM66 in revised manuscript (line 34-49).

As we have added in line 30-33 in revised manuscript, ***“Both PHD finger domain and Bromodomain are evolutionarily and structurally conserved module. The majority of canonical single PHD finger domains were reported to read the N-terminal tail of histone H3, mainly the methylation status of H3K4 (H3K4me2/3 or H3K4me0). While the Bromodomains were found to read the acetylated lysine, especially that on histone H3 and H4.”*** We first perform the GST pulldown assays with calf thymus histones, which showed that TRIM66 PHD-Bromodomain can interact with histones, mainly H3 and H4 (Fig. S2a). According to previous studies, sequence alignment, structure comparisons and the results of GST pulldown, we performed a series of interactional studies between TRIM66 PHD-Bromodomain and histone peptides (unmodified N terminus of H3, H3K14ac, H3K18ac, H3K23ac, H3K27ac, H3K36ac, H3K56ac, H4K5ac, H4K8ac, H4K12ac, H4K16ac, H4K20ac) derived from H3 and H4 by ITC (description was added in line 109-127 in revised manuscript, data was provided in Fig. 1d, S2, S3). We found that TRIM66 PHD-Bromodomain can bind the N terminus of H3 and H3K56ac.

2. Page 6 - To the author's credit, in trying to figure out the TRIM66 code, they learned a lot about the broader TRIM protein family members and the basis for their target recognition. This adds a lot of value to the paper. If it were my paper, I would expand on this and then remove other parts (such as the SIRT6).

Response:

Thanks for your kindly suggestion. After carefully thinking, we tend to keep the part of Sirt6 in our manuscript. Our study not only figured out the specific target recognition of TRIM66 as a “reader” protein, but also focused on the biological function of TRIM66 in the DNA damage response. Our previous results showed that the absence of TRIM66 resulted in invalidated deacetylation of H3K56ac at the DNA damage sites. We believed that the introduction of Sirt6 would be helpful to explore the in-depth mechanism of TRIM66 involved in the H3K56ac deacetylation and DDR in ESCs.

3. There is too much key data in the supplemental figures. This is another symptom of the fact that the paper is too big - needs to be broken up.

Response:

As the reviewer points out, many supplemental results are quite important. We have incorporated

your suggestion to present the data in a reasonable manner. We merged the original Fig. S4 and S5 to the revised Fig. S7. We also deleted the original Fig. S9, which showed the data of adult tissue of TRIM66 deficient mice. And we added new data asked by reviewers in revised Fig. S9, S10, and S11.

4. The NMR data add a lot to this already rigorous manuscript. The technical aspects of combining crystallography, NMR and ITC are outstanding and make for an unusually definitive body of work. This raises the bar for the field.

Response:

Thank you very much for your kind remark. We didn't obtain the complex structure of TRIM66 PHD-Bromodomain and H3K56ac peptide. And the NMR data could provide more details about the interaction between TRIM66 PHD-Bromodomain and H3K56ac peptide.

5. On page 8, regarding aggregation of the peptide-induced protein aggregation: didn't the aggregation problem undermine the experiment? How is the titration interpretable if this is going on? The reader was confused about why this information was in the paper, and whether there were subsequent problems with the data.

Response:

We observed decreasing NMR intensities of some peaks upon the titrations of H3K56ac peptide, this could be ascribed to protein aggregation, diluted concentration caused by the added peptide, intermediate exchange caused by binding of ligand and so on.

To further figure out whether peptide caused protein aggregation or not, we carefully analyzed the linewidth changes of TRIM66-GSGS₉₆₈₋₁₁₆₀ NMR signals along the ¹H dimension, which is sensitive to protein molecular weight changes. To avoid the influence of linewidth broadening caused by the interaction between peptide and protein, we excluded the residues close to the binding interface. And we also excluded the residues located at the flexible region, whose linewidths were insensitive to the changes of molecular weight. We found that many residues located at the rigid region and far from the binding interfaces (e.g., C988, Q1047, L1136, K1048, E1051, L1057, V1108, Y1045, C1050, V975 shown in table below) didn't show obvious linewidth changes (smaller than the digital resolution 5 Hz, experiments were performed on an 850 Hz Bruker NMR spectrometer), indicating that no obvious molecular weight changes happened. These results showed that the addition of H3K56ac peptide didn't induce the aggregation of TRIM66-GSGS₉₆₈₋₁₁₆₀. We have revised the imprecise description to “***Although we observed a modest binding affinity ($K_D = 109.00 \mu M$) in the interaction between the H3K56ac peptide and TRIM66 PHD-Bromodomain (Fig.1d), we observed obvious NMR chemical shift perturbations (Fig.2e and Supplementary Fig.6c) and several peaks even disappeared due to the NMR intermediate exchange effect.***” in line 193-196 in the revised manuscript)

Residue name	Linewidth (Hz) along ¹ H dimension without peptide	Linewidth (Hz) along ¹ H dimension with 1:1.4 peptide
C988	31.5	32.3
Q1047	25	27.7
L1136	21.7	22.5
K1048	26.8	29.9
E1051	36.9	38.8
L1057	28.5	32.8
V1108	26.3	27.4
Y1045	26.4	23.4
C1050	25.1	28.5
V975	35.3	34.6

6. On page 9, given the authors' claims, this reviewer would have liked to see a more direct measure of the upregulated expression of H2AX. The immunoblot is helpful, but it is very qualitative. Other types of direct metrics, along with QRT-PCR would have been important to include.

Response:

Thanks for your suggestions. We further analyzed the expression of a series of genes related to DNA damage via RT-qPCR (Momcilovic et al., 2010; Schmitt et al., 2016; Seifert et al., 2017), and these findings indicated that TRIM66 depletion resulted in the upregulation of DNA damage related genes (*Cdkn1a*, *Gadd45a*, and *p53*) in ESCs (Fig. S9a), consistent with the upregulated expression of γ -H2AX via immunoblot.

Reference:

Momcilovic O, Knobloch L, Fornsgaglio J et al., DNA damage responses in human induced pluripotent stem cells and embryonic stem cells. *PLoS One*, 2010 5, 10: e13410.

Schmitt A, Garcia J, Hung T et al., An inducible long noncoding RNA amplifies DNA damage signaling. *Nat Genet*, 2016 48 (11), 1370-1376.

Seifert B, Dejosez M, and Zwaka T. Ronin influences the DNA damage response in pluripotent stem cells. *Stem Cell Res*, 2017 23, 98-104.

7. In the text, the authors need to be more clear about what is KO-1 and what is KO-2 in Figure 3. The text does not refer effectively to figure 3. In general in this paper, the main text does a poor job referring to the figures, which are not labeled consistently with the language in the text.

Response:

Thanks for the reviewer's suggestions. We have clearly stated the meanings of KO-1 and KO-2 in the corresponding text of Figure 3 and updated the languages in the main text to effectively refer to the figures.

8. *This reviewer is confused by the immunoblot data or perhaps the labeling in Fig. 3 and 5 (as in 5a). The left and right panels differ in the amount of dox exposure (48 vs. 96h), but the control lanes look different under the same conditions (where dox is on in both cases). The layout of these figures was confusing, and the effects did not seem pronounced.*

Response:

Thanks for the reviewer's comments. We have provided more reasonable immunoblot images and clearly labeled according to a previous literature (Zhao et al., 2015) in Fig. 3 and 5 to avoid confusion.

Reference:

Zhao B, Zhang W, Duan Y et al., Filia is an ESC-specific regulator of DNA damage response and safeguards genomic stability. *Cell Stem Cell*, 2015 16, 684-698.

9. *The evidence for an interaction with Sirt6 was not particularly strong. The co-IP could simply illustrate indirect association within a larger complex. Particularly given the biophysical experiments at the beginning of this paper, if the author hopes to stretch the study this far, they should prove a direct interaction between TRIM66 and Sirt6 to make their assertions.*

Response:

Thanks for the reviewer's suggestion. We first tried to test the direct binding of TRIM66₂₄₁₋₉₁₆ with Sirt6 by GST pull-down, but we couldn't obtain the TRIM66₂₄₁₋₉₁₆ protein to do this assay. And then we tried to use yeast two-hybrid system to test the direct binding between TRIM66₂₄₁₋₉₁₆ and Sirt6, but the result was negative. As we couldn't obtain experimental evidence to support the direct interaction now, we speculated that TRIM66 might associate with Sirt6 in a larger complex and corrected our description (in the part "***TRIM66 recruits Sirt6 to deacetylate H3K56ac and safeguards genomic stability in ESCs***") in the revised manuscript to avoid overstatement.

10. *There is too much in this paper - the Sirt6 should be done more rigorously, with some added biophysical binding data, and put into a different paper.*

Response:

Thanks for the reviewer's suggestions. We believed that the introduction of Sirt6 would be helpful to explore the in-depth mechanism of TRIM66 involved in the H3K56ac deacetylation and DDR in ESCs. We also tried to figure out whether the TRIM66 interacted with Sirt6 directly. Unfortunately, we couldn't obtain experimental evidence to support the direct interaction of TRIM66 and Sirt6 now (see the response to comment #9). Thus, we speculated that TRIM66 might associate with Sirt6 in a larger complex and finally decided to keep the part of Sirt6 in our manuscript.

11. *However, this reviewer really liked the data on TRIM66 and murine development (Fig. 7). This was a very nice addition to the paper.*

Response:

We appreciated your agreement on the key roles of Trim66 in murine development. We further derived the primary ESCs from Trim66 KO mice and found that the primary Trim66-depleted ESCs

exhibited elevated levels of γ -H2AX and H3K56ac and a higher rate of chromosomal breakage and chromosome ends fusion (Fig. 7e-h), consistent with the previous results in the isolated blastocysts and mouse ESC line.

Reviewer #3 (Remarks to the Author):

Chen et al determined the crystal structure of the PHD-Bromo tandem domains of TRIM66, free and in complex with unmodified H3 tail. This structural study, together with ITC assays, reveals a specific recognition of the unmodified H3 tail by the PHD-Bromo domains. Through NMR titration, they also identified an interaction between the Bromodomain and the H3(48-57) K56Ac peptide. Their cellular analysis indicated that the TRIM66-histone recognition mediates the K56Ac deacetylation and is important for DNA damage repair and genomic stability in ES cells. Their cellular studies further revealed an interaction between TRIM66 and deacetylase Sirt6, and that such an interaction mediates the deacetylation of H3K56Ac and initiation of DDR. Overall, this study is of interest to understanding the functional role of TRIM66 and the regulatory mechanisms of H3K56Ac and DDR. However, I find that the structure-function connection needs to be strengthened.

1. The authors did not provide any biochemical/cellular evidence to support the claim that “We didn’t find any binding partners of the PHD-Bromodomain from the acetylated H3 and H4 peptides except the acetylated H3K56 peptide.” Such evidence, such as systematic histone array or pull-down assays, would be important for confirming the binding specificity of the Bromodomain.

Response:

As we have added in line 30-33 in the revised manuscript, “**Both PHD finger domain and Bromodomain are evolutionarily and structurally conserved module. The majority of canonical single PHD finger domains were reported to read the N-terminal tail of histone H3, mainly the methylation status of H3K4(H3K4me2/3 or H3K4me0). While the Bromodomains were found to read the acetylated lysine, especially that on histone H3 and H4.**” We first perform the GST pulldown assays with calf thymus histones, which showed that TRIM66 PHD-Bromodomain can interact with histones, mainly H3 and H4 (Fig. S2a). According to previous studies, sequence alignment, structure comparisons and the results of GST pulldown, we performed a series of interactional studies between TRIM66 PHD-Bromodomain and histone peptides (unmodified N terminus of H3, H3K14ac, H3K18ac, H3K23ac, H3K27ac, H3K36ac, H3K56ac, H4K5ac, H4K8ac, H4K12ac, H4K16ac, H4K20ac) derived from H3 and H4 by ITC (description was added in line 109-127, data was provided in Fig. 1d, S2, S3). We found that TRIM66 PHD-Bromodomain can bind the N terminus of H3 and H3K56ac.

Thanks for your kind suggestions, we have modified our description to “**No obvious bindings were observed between TRIM66-WT₉₆₅₋₁₁₆₀ with these acetylated peptides except for the H3₄₈₋₅₇K56ac peptide, which binds to TRIM66-WT₉₆₅₋₁₁₆₀ with a K_D of 109 μ M**” in line 123-125.

2. It is unclear why the authors chose to use the H3(48-57) K56Ac peptide for the binding assay. Given that H3K55 is located at the C-terminal end of the helix 1, one would argue that the binding

assay with the H3(48-57) K56Ac peptide, which mostly likes assumes no secondary structure, would not be sufficient to capture the interaction, if any.

Response:

Previous studies showed that the helix N of H3 might have dynamic nature (Zhang et al., 2018). EPR spectroscopy revealed heterogeneous conformation of the N helix on the H3-H4 tetramer (Bowman et al., 2009). In the structures of some chaperone-H3-H4 complexes, the α N region was invisible (For example, Asf1-H3-H4 (Natsume et al., 2007), Spt2-H3-H4 (Chen et al., 2015), and Mcm2-H3-H4 (Gaubert et al., 2015)).

Reference:

Zhang L, Serra-Cardona A, Zhou, H et al., Multisite Substrate Recognition in Asf1-Dependent Acetylation of Histone H3 K56 by Rtt109. *Cell*. 2018 174, 818-830.e811.

Bowman A, Ward R, El-Mkami H et al., Probing the (H3-H4) 2 histone tetramer structure using pulsed EPR spectroscopy combined with site-directed spin labelling. *Nucleic acids research*. 2009 38, 695-707.

Natsume R, Eitoku M, Akai Y et al., Structure and function of the histone chaperone CIA/ASF1 complexed with histones H3 and H4. *Nature*. 2007 446, 338.

Chen S, Rufiange A, Huang H et al., Structure–function studies of histone H3/H4 tetramer maintenance during transcription by chaperone Spt2. *Genes & Development*. 2015 29, 1326-1340.

Gaubert A, Besle A, Guichard B et al., Structural insight into how the human helicase subunit MCM2 may act as a histone chaperone together with ASF1 at the replication fork. *Nucleic acids research*. 2015 43, 1905-1917.

3. Relatedly, the ITC data for the K56Ac-TRIM66 binding gave a stoichiometric ratio that significantly deviates from 1 (Figure 1d and Figure S3b), which appears to suggest that the observed heat change may not dominantly arise from the specific binding.

Response:

The deviation from a stoichiometric ratio of 1:1 could be caused by the impurities in the synthesized peptide samples, which caused overestimation of peptide concentrations. We improved the peptide quantification method by the quantitative NMR experiments (previously by weight) (Bayle et al., 2015; Huang et al., 2014; Jungnickel and Forbes, 1963; Malz and Jancke, 2005; Simmler et al., 2014; Gao et al., 2014), as aromatic residues were not always present in our synthesized peptides. We have included the following sentences in the method section, now it reads in line 537-538: **“Peptides concentrations were estimated from the quantitative NMR experiments”** We redid the ITC using a MicroCal PEAQ-ITC machine, and the ITC data were subsequently analyzed using the MicroCal PEAQ-ITC analysis software. The ITC data have been updated as now in Fig 1, 4, Fig S2, S3, Table 1 and Table S3. The stoichiometric ratio was now close to 1, e.g., N=1.07 and 1.23 for H3₄₈₋₅₇K56ac peptide binding to TRIM66-WT₉₆₅₋₁₁₆₀ and TRIM66-GSGS₉₆₈₋₁₁₆₀, respectively.

Reference:

Bayle K, Julien M, Remaud G et al., Suppression of radiation damping for high precision quantitative NMR. *J Magn Reson*. 2015 259, 121-125.

Huang T, Zhang W, Dai X et al., Precise measurement for the purity of amino acid and peptide using quantitative nuclear magnetic resonance. *Talanta*. 2014 125, 94-101.

Jungnickel J, and Forbes J. Quantitative Measurement of Hydrogen Types by Intergrated Nuclear Magnetic Resonance Intensities. *Analytical chemistry*. 1963 35, 938-942.

Malz F, and Jancke H. Validation of quantitative NMR. *Journal of Pharmaceutical and Biomedical Analysis*. 2005 38, 813-823.

Simmler C, Napolitano J, McAlpine J et al., Universal quantitative NMR analysis of complex natural samples. *Current Opinion in Biotechnology*. 2014 25, 51-59.

Gao J, Ma R, Wang W et al., Automated NMR fragment based screening identified a novel interface blocker to the LARG/RhoA complex. *PloS one*. 2014 9, e88098.

4. Could the K56Ac-induced TRIM66 aggregation, as the authors proposed, contribute to the heat change?

Response:

We have excluded the possibility of aggregation by linewidth analysis, please see also our response to comment #5 of reviewer #2.

5. How were those peptides containing no Tyr nor Trp quantified for the ITC measurement? Were the errors of the ITC parameters estimated from multiple independent experiments or the curve fitting of one single experiment?

Response:

Since the impurities in the synthesized peptide samples may lead to overestimation of peptide concentrations when quantify the peptide by weight, we determined the peptide concentration using quantitative NMR experiments. Please see also our response to comment #3 as aforementioned.

The errors of the ITC parameters were estimated from the curve fitting of one single experiment. We have clarified it in the footnotes of Table 1 and Table S3.

6. Evidence on a direct interaction between TRIM66 and H3K56ac in cells is also missing. An immunofluorescence assay on the cellular co-localization of H3 K56Ac with TRIM66 wild-type and mutants would help strengthen the link.

Response:

Thanks for your suggestions. To give the evidences on the interaction of TRIM66 and H3K56ac in cells, we performed the immunofluorescence assay in the TRIM66 KO ESCs overexpressed Myc-tagged TRIM66 wild-type or Myc-tagged TRIM66 mutants respectively. Our results showed that only Myc-tagged TRIM66 wild-type was co-localized with H3K56ac perfectly, whereas Myc-tagged TRIM66 mutants were dispersedly distributed in ESCs (Fig. S10). To help strengthen the link, we further provided additional Co-IP data of H3K56ac and TRIM66 wild-type or TRIM66 mutants and these results indicated that H3K56ac interacted with the wild-type TRIM66 rather than these TRIM66 mutants as shown in Fig. 4e.

7. English in the text needs to be improved. There are numerous grammatical issues. To list a few: In the first paragraph, page 2: The sentence “The exact biological function of TRIM66 and its ability to read specific PTMs and execute its biological roles remains unknown” is confusing and needs to be clarified.

In the second paragraph, page 2: Change “are existed” to “exist”.

In the third paragraph, page 2: Change “non-histone proteins, which is...” to “non-histone proteins, which are...”.

In the first paragraph, page 3 and many other places, change “N-terminal of histone H3” to “N-terminus of histone H3”. In the first paragraph, page 5, remove “with” from “contacted with”

In the second paragraph, page 5, change “This finding was consistent” to “This finding is consistent”.

Supplementary Table 5. Change “not detectable binding” to “no detectable binding”.

In Methods, change “The DNA fragments of human TRIM66, TRIM24 and TRIM33” to “The DNA fragment of human TRIM66, TRIM24 and TRIM33”.

In Methods, change “ASF (1-156) into a pRSFDuet” to “ASF (1-156) inserted into a pRSFDuet”.

Response:

Thanks for the reviewer’s suggestions, we have changed this sentence “*The exact biological function of TRIM66 and its ability to read specific PTMs and execute its biological roles remains unknown*” to “**However, little is known about the structure and function of the TRIM66 PHD-bromodomain and it remains a task to figure out whether the PHD-Bromodomain of TRIM66 play a critical role in its cellular function.**” and corrected these issues you mentioned in the revised manuscript and checked the entire contents of the manuscript carefully.

Reviewers' comments:

Reviewer #1 (Remarks to the Author):

The revised manuscript by Chen et al. provides additional support for the role of the bromodomain protein TRIM66 in the DNA damage response. Several new pieces of data strengthen the claim that TRIM66 is involved in genome maintenance. A few previous issues raised by this reviewer have not been well addressed in the revision. It was asked that the authors validate the interaction between wt full-length TRIM66 and the histone modifications from cells. These data were not provided. It was also asked that more direct evidence be provided for how TRIM66 is involved in DNA repair. While there is quite a bit of new data looking at DNA damage signaling and repair following etoposide treatment, these data are not overly convincing. At the dose provided, there are only one or two foci in each cell. The localization of TRIM66 to these sites is provided in New Supp Fig 11 but there is no quantification and the images are not at all convincing. Even when quantification is provided, it does not match the images provided. DNA damaging conditions that induce more damage should be analyzed to see more foci and the potential for TRIM66 to accumulate at these sites. Instead of using repair inhibitors to test the sensitivity of these cells, the sensitivity to DNA damaging agents should be tested. It also appears that all DNA damage signaling and repair are affected which would place TRIM66 at the top of the DSB repair signaling cascade, which is hard to imagine how this would occur in cells. At this point, these data are too speculative to be published in their current form. It would almost be better to take these out than to keep them in the manuscript. While this reviewer believes that this study is interesting and for the most part convincing, the role of TRIM66 in DSB repair is not at all solid enough to be published in this current form.

Reviewer #2 (Remarks to the Author):

The authors have made a reasonable effort to address the comments and enhance the paper.

Reviewer #3 (Remarks to the Author):

In the revised manuscript, the authors have included additional biochemical and cellular data, which significantly strengthened the work and alleviated my concern. I recommend for publication in Nature Communications given the following minor issues to be addressed.

1. Figure 2a. The authors used the same color scheme for the H3 peptide and the PHD finger domain, which can mislead structural interpretation. The position of the label for the PHD domain also needs to be adjusted.
2. Supplementary Figure 1a-c. For clarity, each of the aligned sequences needs to be labelled with actual residual numbers.
3. There is still work to do for text editing. For instance, in page 9, paragraph 2, "structural different of peptide" should be "structural difference of peptide", and "lowest level of binding the complex" should be "lowest level of complex binding".
4. Also in page 9, paragraph 2. The authors reported no observation for the binding between TRIM66 PHD-Bromodomain and nucleosome. A discussion on this result is warranted.

Point-by-point response

Reviewers' comments:

Reviewer #1 (Remarks to the Author):

The revised manuscript by Chen et al. provides additional support for the role of the bromodomain protein TRIM66 in the DNA damage response. Several new pieces of data strengthen the claim that TRIM66 is involved in genome maintenance. A few previous issues raised by this reviewer have not been well addressed in the revision. 1) It was asked that the authors validate the interaction between wt full-length TRIM66 and the histone modifications from cells. These data were not provided. 2) It was also asked that more direct evidence be provided for how TRIM66 is involved in DNA repair. While there is quite a bit of new data looking at DNA damage signaling and repair following etoposide treatment, these data are not overly convincing. At the dose provided, there are only one or two foci in each cell. The localization of TRIM66 to these sites is provided in New Supp Fig 11 but there is no quantification and the images are not at all convincing. Even when quantification is provided, it does not match the images provided. DNA damaging conditions that induce more damage should be analyzed to see more foci and the potential for TRIM66 to accumulate at these sites. 3) Instead of using repair inhibitors to test the sensitivity of these cells, the sensitivity to DNA damaging agents should be tested. It also appears that all DNA damage signaling and repair are affected which would place TRIM66 at the top of the DSB repair signaling cascade, which is hard to imagine how this would occur in cells. At this point, these data are too speculative to be published in their current form. It would almost be better to take these out than to keep them in the manuscript. While this reviewer believes that this study is interesting and for the most part convincing, the role of TRIM66 in DSB repair is not at all solid enough to be published in this current form.

Response:

Thanks for your suggestions. We have further validated the interaction between human TRIM66 and histone peptides using peptide pulldown assay.

The peptide pulldown method: *The HA-tagged TRIM66_{full length} was overexpressed in 293T cells and the N terminal biotinylated peptides (about 1.5 μ l of 10 mM peptide stock solution, which was quantified using quantitative NMR to make sure the different peptides added in equivalent amount) were incubated with 293T cell lysate for 50 min at 4°C (We lysed cells using buffer: 20 mM Tris, 100 mM NaCl, 50 mM KCl, and 1% NP40 at pH 7.5 containing the protease inhibitor (Roche, 05892791001) for 1 h at 4°C, then diluted the cell lysate using buffer without NP40). After 1 h of incubation, 20 μ l streptavidin beads (Thermo, 20359) were washed for five times with wash buffer (20 mM Tris, 100 mM NaCl, 50 mM KCl, and 0.1% NP40 at pH 7.5) and eluted using 2 \times SDS-PAGE sample buffer (0.1 M Tris, 50 mM DTT, 4% SDS, 20% glycerol, and 0.2% Bromophenol blue at pH 6.8), followed by western blot analysis (anti HA-Tag, CST, 3724s, 1:1000; anti-biotin, CST, 7075s, 1:1000).*

The results showed that the Biotin-H3₍₁₋₁₅₎₋₍₄₈₋₅₇₎K56ac peptide displayed a stronger binding ability with full-length wt TRIM66 compared with the Biotin-H3₍₁₋₁₅₎₋₍₄₈₋₅₇₎K56 peptide (the PHD

domain of TRIM66 could bind unmodified H3R2K4) and control (Response Fig. 1a, lane 1-3, up), suggesting that the acetylation of H3K56 recognized by the Bromodomain of TRIM66 might enhance the interaction between full-length *wt* TRIM66 and the peptide. To ensure the equivalent loading of peptides in the peptide pulldown assay, we detected the biotinylated peptides using anti-biotin antibody (Response Fig. 1a, bottom). We found that there were unspecified bands close to the position of biotin-peptides (Response Fig 1a, lane 3, bottom). We further added different amount of biotin-peptides (0 nM, 3.75 nM, 7.5 nM, 12.5 nM, and 15 nM) in the pulldown assay, finding that the unspecified bands were caused by streptavidin beads (Response Fig. 1b, lane 1) and the anti-biotin bands were gradually highlighted with the increasing amount of peptides (Response Fig. 1b, lane 2-5). Considering the anti-biotin bands were similar in peptide pulldown assays (Response Fig. 1a, lane 1 and 2, bottom), we believed the peptide loading was equivalent, which was consistent with our quantitative NMR results.

Response Fig. 1. The interaction between TRIM66 and histone peptides using peptide pulldown assays. **a**, The results of peptide pulldown using Biotin-H3₍₁₋₁₅₎₋₍₄₈₋₅₇₎ K56 and Biotin-H3₍₁₋₁₅₎₋₍₄₈₋₅₇₎ K56ac peptides with full-length *wt* TRIM66 from cells. **b**, Western blot showed the different amount of peptides using anti-biotin antibody.

Further, to help strengthen the link, we provided Co-IP data of H3K56ac and TRIM66 wild-type or TRIM66 mutants and these results indicated that H3K56ac interacted with the wild-type TRIM66 rather than these TRIM66 mutants as shown in Fig. 4e. Then, our results also showed that only Myc-tagged wild-type TRIM66 was co-localized with H3K56ac perfectly, whereas Myc-tagged TRIM66

mutants were dispersedly distributed in ESCs (Fig. S10). These results powerfully supported the interaction between wild-type full-length TRIM66 and histone modifications in ESCs.

In order to further illuminate the role of TRIM66 in the DNA damage repair process of ESCs, we performed cell counting assay and our results showed that TRIM66 depletion significantly impaired the ESC survival under the treatment of Etoposide (EPI), ionizing radiation (IR), or Bleomycin, suggesting that TRIM66 KO ESCs became sensitive to the DNA damage reagents (Fig. S9b). Previous immunofluorescent results showed that Myc-tagged TRIM66 could co-localize with γ -H2AX foci. We further analyzed the co-localization of TRIM66 and γ -H2AX foci under the treatment of DNA damage reagents and our results showed the increased TRIM66 foci, induced by DNA damage reagents, were co-localized with γ -H2AX foci (Fig. S11a and S11b). Notably, WT ESCs displayed a proper competence in DNA damage repair, whereas TRIM66 KO ESCs exhibited an impairment in damage repair (Fig. S11c). These data suggested that TRIM66 was able to bind at the DNA damage sites and participate in the DNA damage repair. Additionally, our in-depth study found that TRIM66 depletion significantly abolished the co-localization of γ -H2AX foci and the DNA repair related proteins (Rad51, CtIP, and DNA-PKcs) rather than the early DNA damage response protein p-ATM in ESCs (Fig. 5f, 5g, S11e, and S11g), suggesting that TRIM66 tend to take part in the later DNA damage repair. Latest literatures verified that Sirt6 was responsible for efficient DNA damage repair (Tian et al., Cell. 2019) and the abnormal deacetylation of H3K56ac affected the DNA damage repair process (Debra et al., Molecular Cell. 2013). Our consistent results also showed that Sirt6 interacted with TRIM66 and was recruited to the genome under the DNA damage stress (Fig. 6c). Taken together, we believed that TRIM66 indeed participated in the DNA damage repair and its interaction with Sirt6 to regulate the H3K56ac modification at the DNA damage sites was crucial for proper DNA repair in ESCs.

[1] Tian X, Firsanov D, Zhang Z, et al., SIRT6 is responsible for more efficient DNA double strand break repair in long-lived species. Cell. 2019. 177(3):622-638.

[2] Debra T, Fabian E, Karim B, et al., SIRT6 recruits SNF2H to DNA break sites, preventing genomic instability through chromatin remodeling. Molecular Cell. 2013. 51:454-468.

Reviewer #2 (Remarks to the Author):

The authors have made a reasonable effort to address the comments and enhance the paper.

Response:

Thank you again for your constructive suggestions and kindly consideration.

Reviewer #3 (Remarks to the Author):

In the revised manuscript, the authors have included addition biochemical and cellular data, which significantly strengthened the work and alleviated my concern. I recommend for publication in Nature Communications given the following minor issues to be addressed.

1. Figure 2a. The authors used the same color scheme for the H3 peptide and the PHD finger

domain, which can mislead structural interpretation. The position of the label for the PHD domain also needs to be adjusted.

2. Supplementary Figure 1a-c. For clarity, each of the aligned sequences needs to be labelled with actual residual numbers.

3. There is still work to do for text editing. For instance, in page 9, paragraph 2, "structural different of peptide" should be "structural difference of peptide", and "lowest level of binding the complex" should be "lowest level of complex binding".

4. Also in page 9, paragraph 2. The authors reported no observation for the binding between TRIM66 PHD-Bromodomain and nucleosome. A discussion on this result is warranted.

Response:

Thanks for the reviewer's suggestions. We have changed the color and the labeling style in the revised manuscript and checked the text editing carefully. And we discussed the observation for the binding between TRIM66 PHD-Bromodomain and nucleosome in the fifth paragraph of Discussion.

REVIEWERS' COMMENTS:

Reviewer #1 (Remarks to the Author):

The revised manuscript by Chen et al. has addressed all of the outstanding issues from previous revisions. The additional data on TRIM66 and DNA damage signaling/repair are greatly improved. Taken together, this manuscript is now suitable for publication in Nature Communications.